# Ursodeoxycholic acid reduces antitumor immunosuppression by inducing CHIP-mediated TGF-β degradation

Yingying Shen[1,12], Chaojie Lu[1,12], Zhengbo Song[2,12], Chenxiao Qiao[1,12], Jiaoli Wang[3,4], Jinbiao Chen[5], Chengyan Zhang[1], Xianchang Zeng[1], Zeyu Ma[1], Tao Chen[6], Xu Li[7], Aifu Lin [8], Jufeng Guo[9], Jianli Wang[10,11,13 ✉] & Zhijian Cai [1,13 ✉]

TGF-β is essential for inducing systemic tumor immunosuppression; thus, blocking TGF-β can greatly enhance antitumor immunity. However, there are still no effective TGF-β inhibitors in clinical use. Here, we show that the clinically approved compound ursodeoxycholic acid (UDCA), by degrading TGF-β, enhances antitumor immunity through restraining Treg cell differentiation and activation in tumor-bearing mice. Furthermore, UDCA synergizes with anti-PD-1 to enhance antitumor immunity and tumor-specific immune memory in tumor-bearing mice. UDCA phosphorylates TGF-β at T282 site via TGR5-cAMP-PKA axis, causing increased binding of TGF-β to carboxyl terminus of Hsc70-interacting protein (CHIP). Then, CHIP ubiquitinates TGF-β at the K315 site, initiating p62-dependent autophagic sorting and subsequent degradation of TGF-β. Notably, results of retrospective analysis shows that combination therapy with anti-PD-1 or anti-PD-L1 and UDCA has better efficacy in tumor patients than anti-PD-1 or anti-PD-L1 alone. Thus, our results show a mechanism for TGF-β regulation and implicate UDCA as a potential TGF-β inhibitor to enhance antitumor immunity.

[1] Institute of Immunology, and Department of Orthopaedics of the Second Affiliated Hospital, Zhejiang University School of Medicine, 310009 Hangzhou, China. [2] Department of Medical Oncology, Zhejiang Cancer Hospital, 310022 Hangzhou, China. [3] Key Laboratory of Clinical Cancer Pharmacology and Toxicology Research of Zhejiang Province, Affiliated Hangzhou First People's Hospital, Zhejiang University School of Medicine, 310006 Hangzhou, China. [4] Zhejiang University Cancer Centre, 310006 Hangzhou, China. [5] Department of Oncology, Hangzhou Xixi Hospital, 310023 Hangzhou, China. [6] Department of Orthopedics, Musculoskeletal Tumor Center, the Second Affiliated Hospital, Zhejiang University School of Medicine, 310003 Hangzhou, China. [7] School of Life Science, Westlake University, 310024 Hangzhou, China. [8] College of Life Sciences, Zhejiang University, 310058 Hangzhou, China. [9] Department of Breast Surgery, Affiliated Hangzhou First People's Hospital, Zhejiang University School of Medicine, 310006 Hangzhou, China. [10] Institute of Immunology, and Bone Marrow Transplantation Center of the First Affiliated Hospital, Zhejiang University School of Medicine, 310058 Hangzhou, China. [11] Institute of Hematology, Zhejiang University & Zhejiang Engineering Laboratory for Stem Cell and Immunotherapy, 310006 Hangzhou, China. [12] These authors contributed equally: Yingying Shen, Chaojie Lu, Zhengbo Song, Chenxiao Qiao. [13] These authors jointly supervised this work: Jianli Wang, Zhijian Cai. ✉email: jlwang@zju.edu.cn; caizj@zju.edu.cn

TGF-β is essential for tumor immunosuppression. After binding and activating the serine/threonine kinase TGF-βRII, TGF-β activates TGF-βRI kinase (ALK5), which phosphorylates Smad2 and Smad3, causing them to interact with Smad4 to form transcription complexes. Activation of TGF-β signaling can diminish antitumor CD4[+] and CD8[+] T cell responses, impair antigen-presenting cell function and increase regulatory T (Treg) cell differentiation[1–3]. TGF-β signaling can be abolished by various mechanisms. Smad7 degrades ALK5 by recruiting the Smurf1/2 and Nedd4-2 E3 ubiquitin ligases[4], dephosphorylates, and inactivates ALK5 by recruiting the phosphatase GADD34-PP1c[5], and prevents Smad2 and Smad3 phosphorylation by forming a stable complex with ALK5[6]. Smad2 and Smad3 can also undergo ubiquitin-mediated degradation by Nedd4L and Smurf1[7]. Although the mechanisms underlying negative regulation of TGF-β signaling have been extensively studied, whether TGF-β signaling can be suppressed through direct degradation of TGF-β is completely unknown.

The development of immune checkpoint inhibitor (ICI) therapy, particularly with antibodies blocking programmed cell death-1/programmed cell death ligand-1 (anti-PD-1/PD-L1 therapy), was a milestone achievement in tumor therapy[8]. After the binding of PD-L1 from the tumor microenvironment (TME)[9], PD-1 on T cells transduces inhibitory signals to impair T cell activation[10,11], which can be blocked by anti-PD-1/PD-L1 therapy. However, many other factors also contribute to the inhibition of T cells. The TME is rich in Treg cells, which are critical for tumor immunosuppression. Notably, PD-1 blockade can accelerate the proliferation and suppressive function of Treg cells, leading to hyperprogressive disease[12]. Given that TGF-β is critical to Treg cell differentiation and function, the combined application of anti-PD-1/PD-L1 therapy and TGF-β pathway inhibitors probably has potent antitumor effects.

Ursodeoxycholic acid (UDCA), a secondary bile acid (BA), is converted from chenodeoxycholic acid by intestinal bacteria[13]. UDCA has long been used in clinical practice for the dissolution of cholesterol gallstones and the treatment of primary biliary cholangitis and other hepatobiliary disorders[14]. Many studies have highlighted the antitumor properties of UDCA. UDCA can reduce the risk of colitis- and chronic liver disease-associated colorectal cancer, induce the apoptosis of various types of tumor cells, and inhibit the proliferation of hepatocellular carcinoma (HCC) cells[15]. However, the effects of UDCA on antitumor immunity are not well defined.

Here, we show that UDCA induces ubiquitination of TGF-β mediated by carboxyl terminus of Hsc70-interacting protein (CHIP), leading to the autophagic sorting and subsequent degradation of TGF-β. In this way, UDCA greatly inhibits the differentiation and activation of Treg cells and eventually attenuates Treg cell-mediated immunosuppression. The combination of UDCA treatment and anti-PD-1 therapy has better antitumor effects than either treatment alone. Thus, our data indicate a mechanism for the posttranslational regulation of TGF-β1 and provide future strategies for tumor therapy.

## Results

**UDCA inhibits tumor progression by modulating Treg cells.**
UDCA substantially inhibited tumor progression in murine models established with B16-F10 melanoma cells, MC38 colon cancer cells, and LLC Lewis lung cancer cells (Fig. 1a) but showed no cytotoxicity in these cells (Supplementary Fig. 1a). Therefore, we investigated whether UDCA can affect antitumor immunity. In UDCA-treated LLC tumor-bearing mice, a decrease in CD4[+] CD25[+] Foxp3[+] Treg cells among tumor-infiltrating leukocytes (TILs) was observed (Fig. 1b). The IL-33 receptor ST2 is critical

for Treg cell production and function[16]. However, UDCA did not alter the ST2 level on Treg cells (Supplementary Fig. 1b). Consistent with the decrease in Treg cells, we detected an increase in CD8[+] T cells among TILs of UDCA-treated tumor-bearing mice (Fig. 1b). We also found that UDCA increased the levels of IFN-γ, perforin, granzyme B and Ki-67 in CD4[+] and CD8[+] T cells and decreased the expression of PD-1 on CD8[+] T cells (Fig. 1b), suggesting the activation of T cells. However, UDCA had no effect on CD19[+] B cells, MHC-II[+] CD11c[+] dendritic cells, F4/80[+] CD11b[+] macrophages, Gr1[+] CD11b[+] neutrophils or CD3⁻ NK1.1[+] NK cells (Supplementary Fig. 1b). Next, we investigated the function of T cells in the UDCA-mediated antitumor effects and found that UDCA did not inhibit tumor growth in NOD Prkdc[scid] Il2rg[−/−] (NSG) mice or athymic nude mice (Supplementary Fig. 1c). Depletion of either CD4[+] T cells or CD8[+] T cells abolished the antitumor effects of UDCA (Supplementary Fig. 1d). Furthermore, depletion of Treg cells was sufficient to abolish the antitumor effects of UDCA (Fig. 1c), which suggested that the increase in CD8[+] T cells was probably caused by a decrease in Treg cells. To confirm this hypothesis, we detected CD8[+] T cells and Treg cells after depletion of Treg cells or CD8[+] T cells, respectively. After Treg cell depletion, UDCA no longer increased the CD8[+] T cell component among TILs (Fig. 1d). However, after the depletion of CD8[+] T cells, UDCA still reduced the Treg cells among TILs (Fig. 1e). Thus, UDCA enhances antitumor immunity by reducing the Treg cell population.

The reduction in Treg cells might stem from decreased induction or proliferation of Treg cells. UDCA did not affect the proportion of Ki-67[+] cells among Treg cells (Supplementary Fig. 1e), indicating inhibition of Treg cell differentiation. TGF-β is essential for Treg cell differentiation[3,17]. Before noticeable differences in tumor size were visible, the levels of the three homologous TGF-β isoforms, TGF-β1, TGF-β2, and TGF-β3, were significantly decreased in the serum and tumor tissues of UDCA-treated mice (Fig. 1f). To investigate the function of TGF-β in the UDCA-mediated reduction in Treg cells, we administered UDCA and found that it did not inhibit tumor growth in LLC tumor-bearing mice with TGF-βRII deficiency (Tgfbr2[f/f]Er-cre, treated with tamoxifen) (Fig. 1g). UDCA neither decreased the Treg cells nor increased the CD8[+] T cell component (Fig. 1h). Similar results were observed in mice deficient for Smad3 (Smad3[−/−]), a central downstream molecule of TGF-β signaling[18,19] (Supplementary Fig. 1f, g). Therefore, UDCA reduced the Treg cells via TGF-β.

TGF-β also contributes to the suppressive function of Treg cells[20], and UDCA significantly reduced the TGF-β level in Treg cells among TILs (Supplementary Fig. 2a). Moreover, UDCA decreased the expression of CTLA4, ICOS, and GITR, the key markers of suppressive function, on Treg cells (Supplementary Fig. 2b). Therefore, we examined the effect of UDCA on Treg cell suppressive function. To obtain enough Treg cells, we analyzed splenic Treg cells and found that UDCA also reduced the population of splenic Treg cells and their TGF-β1, CTLA4, ICOS, and GITR levels in tumor-bearing mice (Supplementary Fig. 2c–e). Therefore, we evaluated the inhibitory capacity of splenic Treg cells and found that it was significantly decreased by UDCA (Fig. 1i). However, UDCA did not affect the inhibitory capacity of splenic Treg cells from Tgfbr2[f/f]Er-cre tumor-bearing mice (Supplementary Fig. 2f). Collectively, these results indicate that UDCA enhances antitumor immunity by inhibiting the differentiation and activation of Treg cells in a TGF-β-dependent manner.

**UDCA inhibits Treg cell induction and function by reducing TGF-β.** To further investigate the function of UDCA in Treg cell

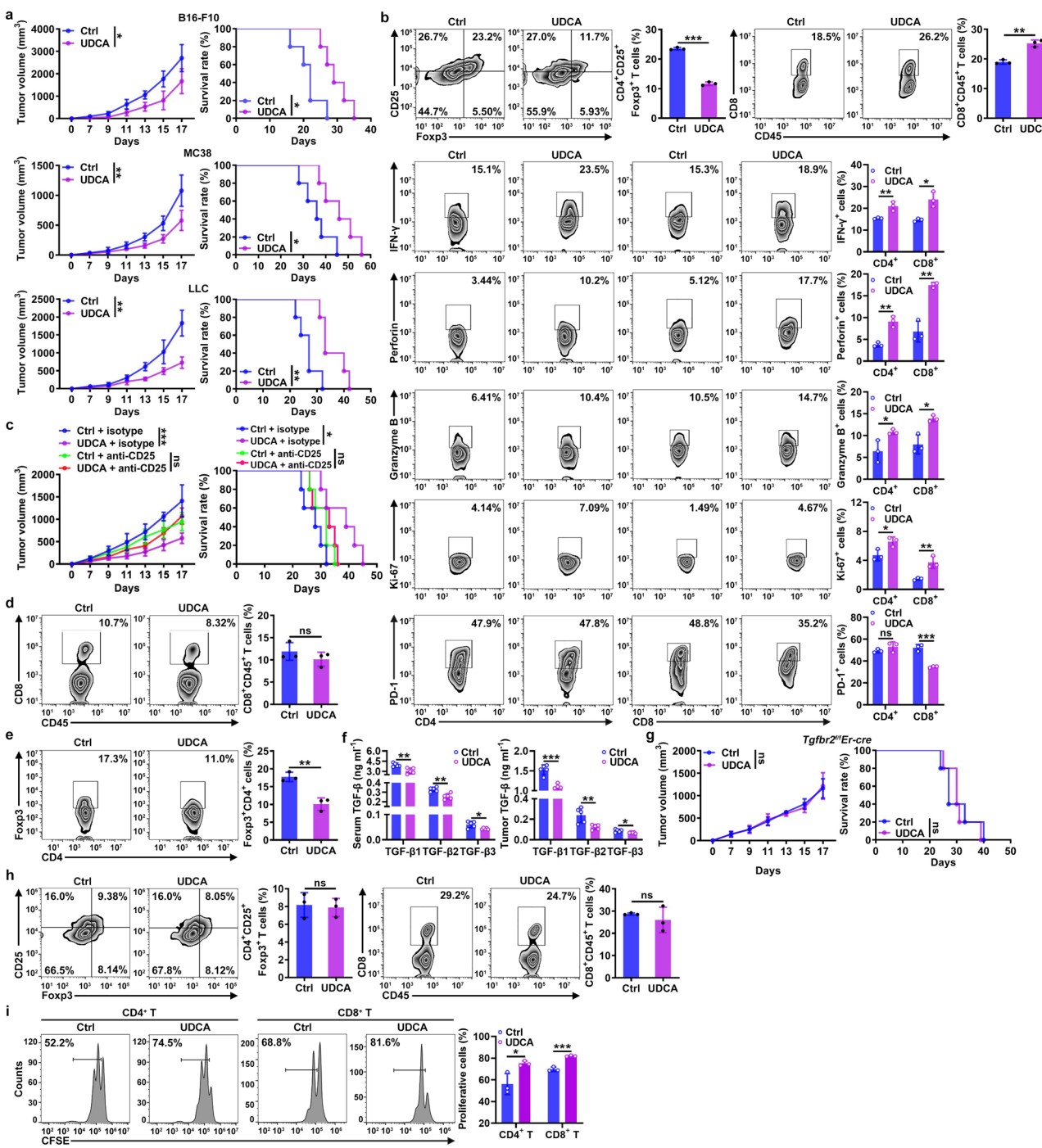

**Fig. 1 UDCA inhibits tumor progression by modulating Treg cells. a, b** Tumor sizes and mouse survival (**a**) and flow cytometric (FC) analysis of CD4+CD25+Foxp3+ Treg cells, total CD8+ T cells and the corresponding CD4+ and CD8+ T cell subsets among TILs on day 18 (**b**) in B16-F10, MC38 (**a**) and LLC (**a, b**) tumor-bearing mice that received intraperitoneal (i.p.) injection of 30 mg kg$^{-1}$ UDCA every 2 days. **c–e** Tumor sizes and mouse survival (**c**) and FC analysis of CD8+CD45+ T cells and Foxp3+CD4+CD45+ T cells among TILs on day 18 (**d, e**) in LLC tumor-bearing mice treated with UDCA along with 100 μg of anti-CD25 (**c, d**) or 40 μg of anti-CD8-neutralizing antibodies (anti-CD8) (**e**) every 3 days. **f** ELISA of TGF-β1, TGF-β2, and TGF-β3 in serum and tumor tissues of UDCA-treated LLC tumor-bearing mice on day 9. **g, h** Tumor sizes and mouse survival (**g**) and FC analysis of CD4+CD25+Foxp3+ Treg cells and CD8+CD45+ T cells among TILs on day 18 (**h**) in LLC tumor-bearing *Tgfbr2f/fEr-cre* mice treated with UDCA. **i** FC analysis of the proliferation of CFSE-labeled CD4+ or CD8+ T cells cocultured with CD4+CD25+ Treg cells isolated from the spleens of UDCA-treated LLC tumor-bearing mice on day 18. Cells were cultured at a T cell:Treg cell ratio of 4:1 in anti-CD3 and anti-CD28-coated plates for 5 days. Representative results from three independent experiments are shown (*n* = 5 in (**a, c, f, g**); *n* = 3 in (**b, d, e, h, i**). \**P* < 0.05; \*\**P* < 0.01; \*\*\**P* < 0.001; ns, not significant (unpaired two-tailed Student's *t* test except for log-rank test for survival rate analysis; mean and s.d.). See Source Data file for the exact *P*-values.

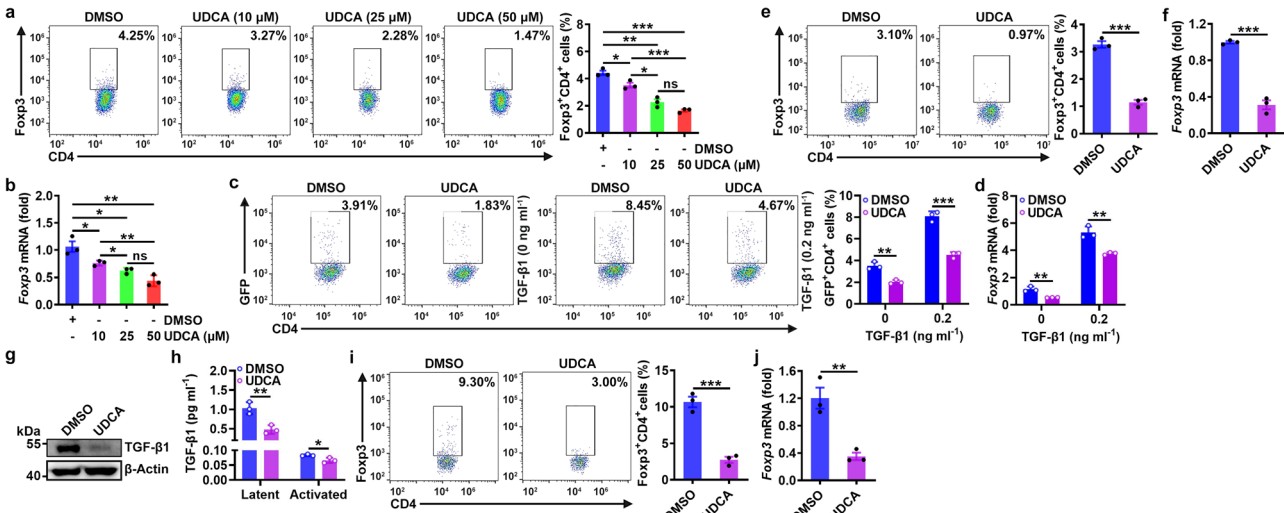

**Fig. 2 UDCA inhibits Treg cell induction and function by reducing the TGF-β level. a**, **b** FC analysis of Foxp3+CD4+ T cells (**a**) and real-time PCR analysis of *Foxp3* mRNA expression (**b**) in naïve CD4+ T cells stimulated with anti-CD3, anti-CD28, and the indicated concentration of UDCA for 3 days. **c**, **d** FC analysis of GFP+CD4+ T cells (**c**) and real-time PCR analysis of *Foxp3* mRNA expression (**d**) in naïve CD4+ T cells from *Foxp3GFP* mice stimulated with anti-CD3, anti-CD28, and 50 μM UDCA in the absence or presence of the cytokine TGF-β1 (0.2 ng ml⁻¹) for 3 days. **e**, **f** FC analysis of Foxp3+CD4+ T cells (**e**) and real-time PCR analysis of *Foxp3* mRNA expression (**f**) in naïve CD4+ T cells from OT-II mice stimulated with 2 μg ml⁻¹ OVA$_{323-339}$ peptide plus T cell-depleted γ-irradiated splenic cells in the presence of 50 μM UDCA for 3 days. **g**, **h** Immunoblot (IB) analysis (**g**) or ELISA (**h**) of TGF-β1 in naïve CD4+ T cells stimulated with anti-CD3, anti-CD28, and 50 μM UDCA for 24 h (**g**) or in the culture supernatant (**h**). **i**, **j** FC analysis of Foxp3+CD4+ T cells (**i**) and real-time PCR analysis of *Foxp3* mRNA expression (**j**) in naïve CD4+ T cells from OT-II mice cocultured with LLC-OVA cells in the presence of 50 μM UDCA for 3 days. Representative results from two (**b**, **d**, **f**, **j**) or three (**a**, **c**, **e**, **g–i**) independent experiments are shown ($n = 3$ in all statistical groups). *$P < 0.05$; **$P < 0.01$; ***$P < 0.001$; ns, not significant (unpaired two-tailed Student's *t* test; mean and s.d.). See Source Data file for the exact *P*-values.

differentiation, we differentiated naïve CD4+ T cells into Treg cells by T cell receptor (TCR) stimulation combined with TGF-β1 cytokine treatment. We found that at high concentrations of TGF-β1 (1 and 10 ng ml⁻¹), UDCA did not affect Treg cell differentiation. However, at low concentrations of TGF-β1 (0.01 and 0.2 ng ml⁻¹), UDCA greatly inhibited Treg cell differentiation (Supplementary Fig. 3a). Therefore, we assumed that UDCA might suppress Treg cell differentiation by inhibiting the expression of endogenous TGF-β1, which is abolished by the presence of excessive exogenous TGF-β1. Actually, without exogenous TGF-β1, UDCA dose-dependently suppressed the TCR stimulation-driven induction of Treg cell differentiation and *Foxp3* mRNA expression (Fig. 2a, b). Similarly, UDCA inhibited Treg cell induction in TCR-stimulated naïve CD4+ T cells from *Foxp3GFP* transgenic mice in both the presence and absence of 0.2 ng/ml exogenous TGF-β1 (Fig. 2c, d). The effect of TGF-β in serum was excluded because UDCA still inhibited Treg cell differentiation in serum-free medium (Supplementary Fig. 3b). The effects of UDCA on T cell proliferation and apoptosis were also excluded (Supplementary Fig. 3c, d). In addition to UDCA, other types of BAs also suppressed TCR stimulation-driven Treg cell generation (Supplementary Fig. 3e). We extended this finding to antigen-specific stimulation and found that UDCA impeded the differentiation of naïve CD4+ T cells from OT-II transgenic mice into Treg cells in response to their cognate antigen OVA$_{323-339}$ (Fig. 2e, f). Then, we evaluated the effects of UDCA on TGF-β1 expression in TCR-stimulated CD4+ T cells. UDCA substantially reduced TGF-β1 protein but not mRNA expression in TCR-stimulated CD4+ T cells (Fig. 2g and Supplementary Fig. 3f). Furthermore, UDCA reduced the protein levels of both latent and activated TGF-β1 in the supernatant of TCR-stimulated CD4+ T cells (Fig. 2h). Along with the decrease in TGF-β1 levels, the level of phosphorylated Smad3 was also decreased (Supplementary Fig. 3g). Accordingly, UDCA inhibited the expression of the TGF-β-inducible *Smad7* and *Fos* mRNAs[21,22] (Supplementary

Fig. 3h). However, UDCA did not affect the mRNA expression of *Tgfbr1* or *Tgfbr2* (Supplementary Fig. 3i). D-mannose induces Treg cells by activating latent TGF-β1[23]. To exclude the possibility that UDCA suppresses Treg cell induction by reducing latent TGF-β1 activation, we induced differentiation and found that UDCA did not affect Treg cell induction in the presence of latent TGF-β1 (Supplementary Fig. 3j). Thus, we concluded that UDCA suppresses the activation of TGF-β signaling by directly reducing TGF-β1 expression.

We next examined the function of TGF-β in UDCA-mediated inhibition of Treg cell induction. TCR-stimulated CD4+ T cells generated a similar number of Treg cells in the presence of UDCA in combination with an ALK5 inhibitor (Supplementary Fig. 3k). Similar results were observed in TCR-stimulated CD4+ T cells from *Tgfbr2f/f Er-cre* mice (Supplementary Fig. 3l). These data indicate that UDCA inhibits Treg cell induction in a manner dependent on TGF-β. Cytokine cues that naïve CD4+ T cells encounter in vivo dictate their differentiation fate[24]. Tumor cells are the main TGF-β source in the TME[25]. We then tested whether UDCA can inhibit Treg cell differentiation by reducing TGF-β secretion from tumor cells. First, we confirmed that UDCA notably suppressed the expression of all TGF-β isoforms in OVA-expressing LLC cells (LLC-OVA cells) (Supplementary Fig. 3m). After coculture with LLC-OVA cells in a medium with or without serum, numerous naïve CD4+ T cells from OT-II mice differentiated into Treg cells, but this differentiation was significantly inhibited by UDCA (Fig. 2i, j and Supplementary Fig. 3n). Then, we investigated whether UDCA affects the function of Treg cells induced by culture with tumor cell supernatant. In line with the in vivo results, culture with supernatant from UDCA-treated LLC-OVA cells induced Treg cells expressing reduced levels of TGF-β, CTLA4, ICOS, and GITR (Supplementary Fig. 3o). Accordingly, the suppressive capacity of these Treg cells was also reduced (Supplementary Fig. 3p). Taken together, these findings indicate that UDCA

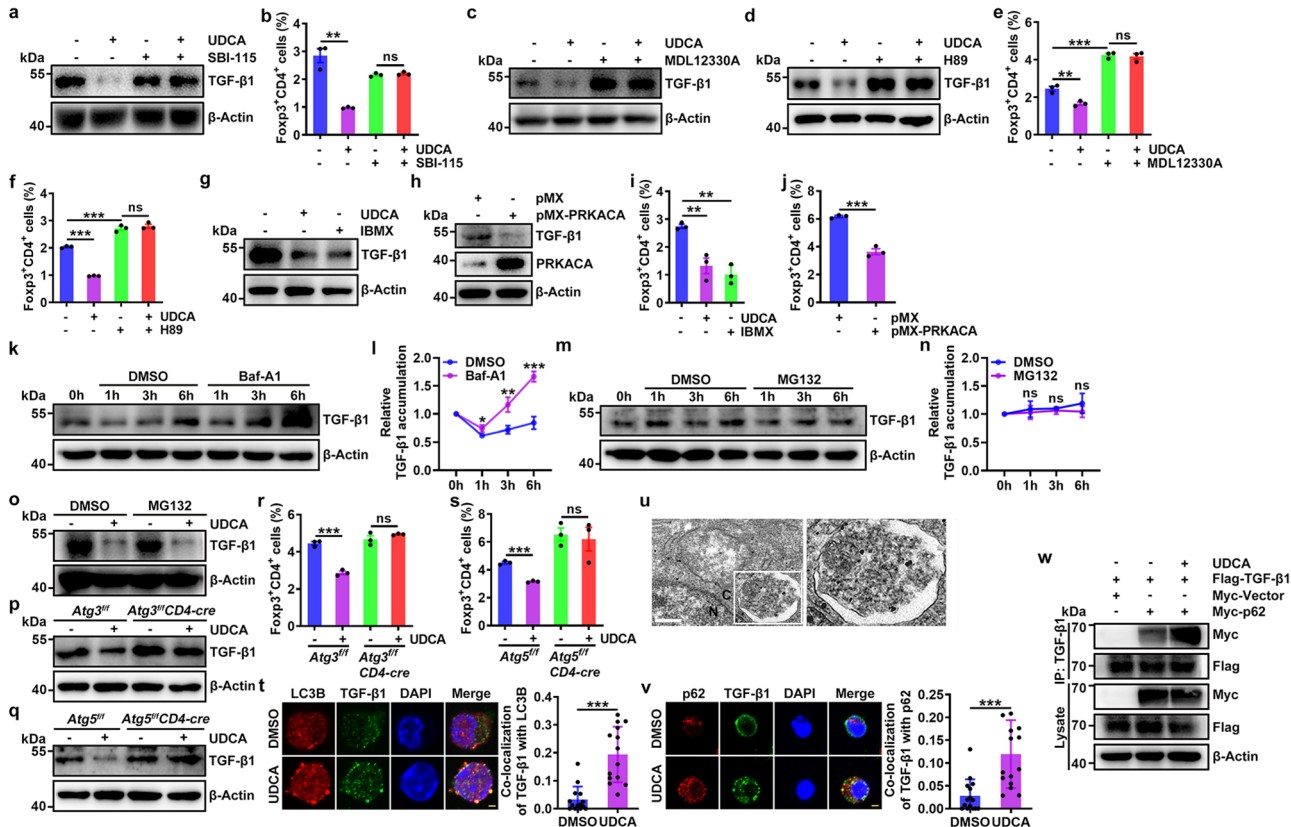

**Fig. 3 UDCA induces p62-dependent autophagic degradation of TGF-β. a–j** IB analysis of TGF-β1 (**a, c, d, g, h**) and FC analysis of Foxp3+CD4+ T cells (**b, e, f, i, j**) in naïve CD4+ T cells treated with 0.25 μM SBI-115 (**a, b**), 5 μM MDL12330A (**c, e**), 2 μM H89 (**d, f**), or 10 μM IBMX (**g, i**) or in naïve CD4+ T cells with PRKACA overexpression (**h, j**) and treated with anti-CD3, anti-CD28, and 50 μM UDCA for 24 h (**a, c, d, g, h**) or for 4 days (**b, e, f, i, j**). **k–n** IB analysis of TGF-β1 (**k, m**) and statistical analysis of TGF-β1 stability (**l, n**) in naïve CD4+ T cells stimulated with anti-CD3, anti-CD28, 10 nM Baf-A1 (**k, l**) or 20 μM MG132 (**m, n**) for the indicated times. **o–s** IB analysis of TGF-β1 (**o–q**) and FC analysis of Foxp3+CD4+ T cells (**r, s**) in WT (**o**), Atg3-deficient or Atg5-deficient (**p, q**) naïve CD4+ T cells stimulated with anti-CD3, anti-CD28, or 50 μM UDCA with (**o**) or without (**p, q**) MG132 treatment for 24 h (**o–q**) or for 4 days (**r, s**). **t** Immunofluorescence analysis of LC3B and TGF-β1 in naïve CD4+ T cells stimulated with anti-CD3, anti-CD28, and 50 μM UDCA for 24 h. Scale bar, 2 μm. **u** Immunoelectron microscopy analysis of TGF-β1 localization in autophagosomes of HEK293T cells. N, nucleus; C, cytoplasm. Scale bar, 500 nm. **v** Immunofluorescence analysis of p62 and TGF-β1 in naïve CD4+ T cells stimulated with anti-CD3, anti-CD28, and 50 μM UDCA for 24 h. Scale bar, 2 μm. **w** IB analysis of p62 and TGF-β1 in lysates of HEK293T cells transfected with vectors expressing Flag-TGF-β1 and Myc-p62 with or without UDCA and 1 nM Baf-A1 treatment for 24 h. IB analysis was conducted after IP with anti-Flag. Representative results from two (**t, v**) or three (**a–s, u, w**) independent experiments are shown (n = 3 in **b, e, f, i, j, l, n, r, s**; n = 14 in **t**; n = 15 in **v**). *P < 0.05; **P < 0.01; ***P < 0.001; ns, not significant (unpaired two-tailed Student's t test except for one-way ANOVA followed by Newman–Keuls test in (**e, f, i**); mean and s.d.). See Source Data file for the exact P-values.

inhibits Treg cell differentiation and activation by reducing the TGF-β level.

**UDCA induces p62-dependent autophagic degradation of TGF-β.** Next, we investigated how UDCA reduces the TGF-β level. TGR5 is a membrane receptor activated by BAs, and farnesoid X receptor (FXR) is a nuclear receptor dedicated to BA signaling[26]. We examined the function of these two receptors in the UDCA-mediated reduction in the TGF-β level and found that the TGR5 agonist INT777 but not the FXR agonist INT747 decreased the TGF-β1 protein level in CD4+ T cells and inhibited Treg cell differentiation (Supplementary Fig. 4a, b). Furthermore, the TGR5 inhibitor SBI-115 completely abolished the UDCA-mediated reduction in the TGF-β1 protein level and Treg cell differentiation (Fig. 3a, b). The binding of TGR5 activates adenylate cyclase, resulting in the elevation of cyclic AMP (cAMP) and subsequent activation of protein kinase A (PKA)[26]. UDCA treatment indeed increased the cAMP level and activated PKA in CD4+ T cells (Supplementary Fig. 4c, d). In contrast to UDCA, the adenylate cyclase inhibitor MDL12330A or the selective PKA inhibitor H89 markedly increased the TGF-β1 level in CD4+

T cells and promoted Treg cell differentiation (Fig. 3c–f). In addition, both MDL12330A and H89 abolished the UDCA-mediated reductions in the TGF-β1 level and Treg cell differentiation (Fig. 3c–f). In contrast, a decrease in cAMP hydrolysis mediated by the phosphodiesterase inhibitor isobutylmethylxanthine (IBMX) or overexpression of cAMP-dependent protein kinase catalytic subunit α (PRKACA) was sufficient to reduce the TGF-β1 level and Treg cell differentiation (Fig. 3g–j). These results indicate that UDCA reduces the TGF-β level via the TGR5-cAMP-PKA axis.

UDCA did not change *Tgfb1* mRNA expression in TCR-stimulated CD4+ T cells (Supplementary Fig. 3f), and in the presence of cycloheximide (CHX), UDCA still reduced the TGF-β level (Supplementary Fig. 4e). Therefore, we assumed that UDCA probably affects TGF-β1 degradation. The proteasome system and autophagy are the major protein degradation pathways. The autophagy inhibitor bafilomycin A1 (Baf-A1) but not the proteasome inhibitor MG132 markedly increased TGF-β1 accumulation in CD4+ T cells (Fig. 3k–n); this result was further confirmed in *Atg3*−/− and *Atg5*−/− CD4+ T cells from *Atg3f/f CD4-cre* and *Atg5f/fCD4-cre* mice (Supplementary Fig. 4f–j). Accordingly, UDCA still reduced the TGF-β1 level in MG132-

treated CD4[+] T cells (Fig. 3o). However, UDCA no longer reduced the TGF-β1 level or inhibited Treg cell differentiation in either *Atg3*[−/−] or *Atg5*[−/−] CD4[+] T cells or in Baf-A1-treated wild-type (WT) CD4[+] T cells (Fig. 3p–s and Supplementary Fig. 4k, l). In addition, UDCA reduced exogenous TGF-β1 expression in HEK293T cells, which was also abolished by Baf-A1 (Supplementary Fig. 4m). Moreover, UDCA did not reduce the TGF-β2 or TGF-β3 level in Baf-A1-treated CD4[+] T cells (Supplementary Fig. 4n). These results indicate that UDCA promotes TGF-β degradation via the autophagy pathway.

Since UDCA did not induce autophagy (Supplementary Fig. 5a, b), we hypothesized that the enhanced TGF-β degradation mediated by UDCA probably stems from increased localization of TGF-β in autophagosomes. UDCA indeed increased the localization of TGF-β1 in autophagosomes in TCR-stimulated CD4[+] T and HEK293T cells (Fig. 3t and Supplementary Fig. 5c). Immunoelectron microscopy further confirmed the distribution of TGF-β1 in autophagosomes in HEK293T cells (Fig. 3u). This effect was cAMP- and PKA-dependent, because in the presence of MDL12330A or H89, UDCA no longer induced the accumulation of TGF-β1 in autophagosomes (Supplementary Fig. 5d, e). In contrast, IBMX treatment or PRKACA overexpression alone was sufficient to promote autophagosomal localization of TGF-β1 in CD4[+] T cells or NIH-3T3 human fibroblasts, respectively (Supplementary Fig. 5f, g).

Adaptor proteins, including p62, NBR1, NDP52, TAX1BP1, and Optineurin, are responsible for the sorting of proteins into autophagosomes[27]. UDCA may increase the autophagosomal localization of TGF-β by promoting its interaction with one or more adaptor proteins. First, we screened for the adaptor protein or proteins that determine UDCA-mediated TGF-β degradation. Knockdown of p62 but not other adaptor proteins abrogated UDCA-mediated degradation of exogenous TGF-β1 in NIH-3T3 cells (Supplementary Fig. 5h). p62 sorts proteins into autophagosomes by directly binding to LC3B[28]. The proximity ligation assay (PLA) results suggested that UDCA induced the interaction of TGF-β1 and LC3B via p62 (Supplementary Fig. 5i). In addition, p62 silencing abolished UDCA-mediated TGF-β1 degradation in CD4[+] T cells and UDCA-mediated inhibition of Treg cell differentiation (Supplementary Fig. 5j, k). Then, we examined the effect of UDCA on the TGF-β1–p62 interaction and found that UDCA notably induced this interaction (Fig. 3v). Moreover, UDCA obviously induced the co-immunoprecipitation (co-IP) of TGF-β1 by p62 (Fig. 3w). Similarly, in the presence of MDL12330A or H89, UDCA did not induce the interaction of TGF-β1 and p62 (Supplementary Fig. 5l, m). In contrast, IBMX treatment or PRKACA overexpression notably promoted this interaction (Supplementary Fig. 5n, o). Collectively, these results indicate that UDCA promotes the p62-dependent autophagy aggregation and subsequent degradation of TGF-β1 via the TGR5-cAMP-PKA axis.

**UDCA promotes TGF-β1 ubiquitination at K315 by CHIP.** p62 binds to ubiquitinated proteins via a ubiquitin-associated (UBA) domain[28]. Full-length p62 (p62_FL) but not p62 lacking the UBA domain (p62_UBAΔ) interacted with TGF-β1 (Supplementary Fig. 6a). Therefore, UDCA probably enhances the interaction of TGF-β1 and p62 by promoting TGF-β1 ubiquitination. We found that UDCA significantly increased total and K63-linked ubiquitination but not K48-linked ubiquitination of TGF-β1 in TCR-stimulated CD4[+] T cells (Fig. 4a). Furthermore, PRKACA overexpression increased total and K63-linked ubiquitination of TGF-β1 in HEK293T cells (Supplementary Fig. 6b, c). These results suggest that the E3 ubiquitin ligase responsible for TGF-β1 ubiquitination probably interacts with both TGF-β1 and PKA.

Then, we analyzed protein interactions with TGF-β1 and PKA by mass spectrometry (MS). There were 14 E3 ubiquitin ligases interacting with TGF-β1 and PKA, respectively (Supplementary Fig. 6d). Among these E3 ligases, 7 interacted with both TGF-β1 and PKA (Supplementary Fig. 6d). We next knocked down these ligases individually and found that knockdown of CHIP but not the other 6 E3 ligases completely blocked UDCA-induced TGF-β1 degradation in NIH-3T3 cells (Supplementary Fig. 6e). Moreover, CHIP knockdown blocked PKA-mediated TGF-β1 degradation in NIH-3T3 cells (Supplementary Fig. 6f). In contrast, CHIP overexpression notably reduced the amount of TGF-β1 protein but not mRNA in HEK293T cells (Supplementary Fig. 6g, h). Then, we confirmed the interaction of CHIP and TGF-β1 (Fig. 4b) and found that CHIP silencing abolished the UDCA-mediated reduction in the TGF-β1 level and inhibition of Treg cell induction (Fig. 4c, d). Consistent with this finding, the autophagosomal localization of TGF-β1 was notably reduced after CHIP silencing (Fig. 4e). To further show that CHIP is an E3 ubiquitin ligase of TGF-β1, we performed in vitro ubiquitination assays and found that CHIP directly ubiquitinated TGF-β1 (Fig. 4f). Furthermore, UDCA resulted in CHIP-dependent degradation of TGF-β2 and TGF-β3 (Supplementary Fig. 6i). Thus, we concluded that CHIP is an E3 ubiquitin ligase of TGF-β.

Then, we sought to identify the ubiquitinated site(s) in TGF-β1. We mutated 20 lysines in TGF-β1 individually and constructed vectors expressing the corresponding TGF-β1 mutants. We found that PKA-induced ubiquitination of the TGF-β1K315R mutant was abolished (Supplementary Fig. 6j) and that CHIP could not ubiquitinate the TGF-β1K315R mutant in in vitro ubiquitination assays (Fig. 4g). Furthermore, neither UDCA treatment nor PRKACA overexpression was able to induce the p62 interaction or autophagosomal accumulation of the TGF-β1K315R mutant (Fig. 4h–k). Collectively, these results demonstrate that CHIP, functioning as an E3 ubiquitin ligase, directly ubiquitinates TGF-β1 at K315.

**TGF-β1 with T282 phosphorylation by PKA binds increased CHIP.** Then, we investigated how PKA promotes CHIP-mediated TGF-β1 ubiquitination. We found that PRKACA overexpression did not induce Ser/Thr phosphorylation of CHIP but markedly promoted that of TGF-β1 in HEK293T cells (Supplementary Fig. 7a, b). In addition, UDCA treatment increased Ser/Thr phosphorylation of TGF-β1 in a PKA-dependent manner in CD4[+] T cells (Fig. 5a). In addition, the interaction of TGF-β1 and PRKACA was confirmed in HEK293T cells (Supplementary Fig. 7c). Then, we confirmed that recombinant PRKACA directly phosphorylated TGF-β1 in an in vitro kinase assay (Fig. 5b). Given that neither UDCA treatment nor overexpression of PRKACA increased the CHIP protein level in HEK293T cells (Supplementary Fig. 7d, e), we hypothesized that PKA-induced phosphorylation of TGF-β1 may enhance its interaction with CHIP and its subsequent ubiquitination. As expected, PRKACA overexpression greatly increased the interaction of CHIP and TGF-β1 in HEK293T cells (Supplementary Fig. 7f). In addition, UDCA increased the interaction of CHIP and TGF-β1 via PKA in CD4[+] T cells (Fig. 5c). Then, we identified the phosphorylated site(s) in TGF-β1. The MS results indicated phosphorylation of TGF-β1 at T282, which is conserved in mammals (Supplementary Fig. 7g). Then, we constructed a vector expressing a TGF-β1 mutant in which T282 was replaced with alanine. We confirmed that PRKACA overexpression-induced TGF-β1 phosphorylation of the TGF-β1T282A mutant was abolished (Fig. 5d). In addition, the results of the in vitro kinase assay suggested that almost no phosphorylation of the TGF-β1T282A mutant by recombinant PRKACA occurred (Fig. 5e). A pulldown (PD) assay demonstrated that the TGF-

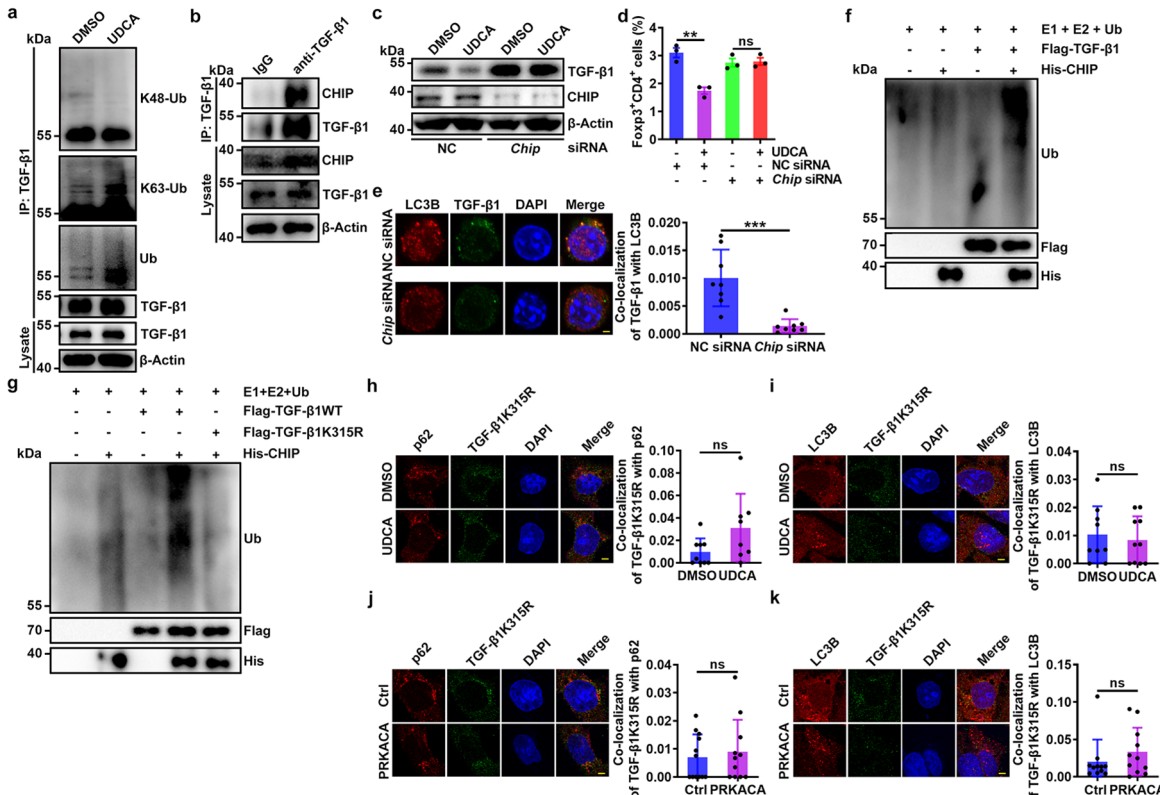

**Fig. 4 UDCA promotes TGF-β1 ubiquitination at K315 by CHIP. a** IB analysis of K48-linked, K63-linked and total ubiquitination of TGF-β1 in naïve CD4+ T cells stimulated with anti-CD3, anti-CD28, 50 μM UDCA and 1 nM Baf-A1 for 24 h. IB analysis was conducted after IP with anti-TGF-β1. **b** IB analysis of CHIP and TGF-β1 in naïve CD4+ T cells stimulated with anti-CD3 and anti-CD28 for 24 h. IB analysis was conducted after IP with anti-TGF-β1 or IgG. **c–e** IB analysis of TGF-β1 (**c**), FC analysis of Foxp3+CD4+ cells (**d**) and immunofluorescence analysis of LC3B and TGF-β1. Scale bar, 2 μm (**e**) in CHIP-knockdown naïve CD4+ T cells stimulated with anti-CD3, anti-CD28 and 50 μM UDCA for 24 h (**c, e**) or for 4 days (**d**). **f, g** IB analysis of TGF-β1 ubiquitination in the presence of E1, E2, Ub, purified CHIP, and TGF-β1 (**f**) or TGF-β1K315R (**g**). **h–k** Immunofluorescence analysis of p62 and TGF-β1K315R (**h, j**) or LC3B and TGF-β1K315R (**i, k**) in NIH-3T3 cells treated with 50 μM UDCA (**h, i**) or transfected with vectors expressing PRKACA (**j, k**) and transfected with the TGF-β1K315R mutant plasmid for 24 h. Scale bar, 2 μm. Representative results from two (**e, h–k**) or three (**a–d, f, g**) independent experiments are shown [n = 3 in (**d**); n = 8 in (**e**); n = 9 (DMSO) or 8 (UDCA) in (**h**); n = 9 (DMSO) or 10 (UDCA) in (**i**); n = 12 (Ctrl) or 8 (PRKACA) in (**j**); n = 11 in (**k**)]. **P < 0.01; ***P < 0.001; ns, not significant (unpaired two-tailed Student's t test; mean and s.d.). See Source Data file for the exact P-values.

β1T282A mutant barely interacted with CHIP (Fig. 5f). Correspondingly, CHIP overexpression had almost no effect on enhancing the ubiquitination and degradation of the TGF-β1T282A mutant (Supplementary Fig. 7h, i). In addition, in vitro ubiquitination assays demonstrated that recombinant CHIP barely ubiquitinated the TGF-β1T282A mutant (Fig. 5g). To further investigate the functional relevance of TGF-β1 T282, we transfected TGF-β1T282AWT or the TGF-β1T282A mutant into HEK293 cells and found that the TGF-β1T282A mutant had almost no autophagosomal localization, while TGF-β1T282AWT had slight autophagosomal localization (Fig. 5h). Correspondingly, culture with supernatant from cells expressing the TGF-β1T282A mutant increased Treg cell induction (Fig. 5i). Collectively, these results indicate that PKA promotes the ubiquitination and autophagosomal localization of TGF-β1 by phosphorylating it at the T282 site.

**CHIP inhibits tumor progression by reducing TGF-β level.** Then, we investigated the functions of CHIP in tumor progression. First, we confirmed that UDCA inhibited the differentiation of human Treg cells in a TGF-β1-dependent manner (Fig. 6a). Then, we found that the TGF-β1, TGF-β2, and TGF-β3 levels in the supernatants of A549 human lung cancer cells and SW480 human colon cancer cells decreased after UDCA treatment; in addition, the differentiation of human Treg cells induced by the supernatants of both UDCA-treated cells was reduced, and this

effect was abolished by TGF-β1-neutralizing antibodies (anti-TGF-β1) (Fig. 6b, c). Consistent with the reductions in the TGF-β1, CTLA4, ICOS, and GITR levels, the suppressive capacity of Treg cells induced by A549 cell supernatant was also reduced by UDCA (Supplementary Fig. 8a, b). Given that UDCA promotes the degradation of TGF-β by CHIP, we analyzed the correlation between TGF-β1 and CHIP protein levels in tumor tissues of non-small-cell lung cancer (NSCLC) patients (Supplementary Table 1). Compared to those in paracancerous (Para-Ca) tissues, CHIP protein levels in cancerous (Ca) tissues were significantly decreased (Fig. 6d). In contrast, TGF-β1 protein levels in Ca tissues were significantly enhanced (Fig. 6e). As expected, the protein levels of CHIP and TGF-β1 were negatively associated (Fig. 6f). In addition, the levels of phosphorylated PKA were significantly reduced in Ca tissues and negatively correlated with TGF-β1 levels (Fig. 6g, h). However, the protein levels of phosphorylated PKA and CHIP exhibited no correlation (Supplementary Fig. 8c). Furthermore, unfavorable progression-free survival (PFS) was observed in NSCLC patients with lower CHIP protein levels (Fig. 6i). The abundant TGF-β in the HCC microenvironment has unique effect on HCC development[29]. Data from the CPTAC database demonstrated that TGF-β1 and CHIP protein levels were also negatively correlated in patients with HCC (Fig. 6j). Collectively, these results suggest that accelerated tumor progression is related to downregulation of CHIP.

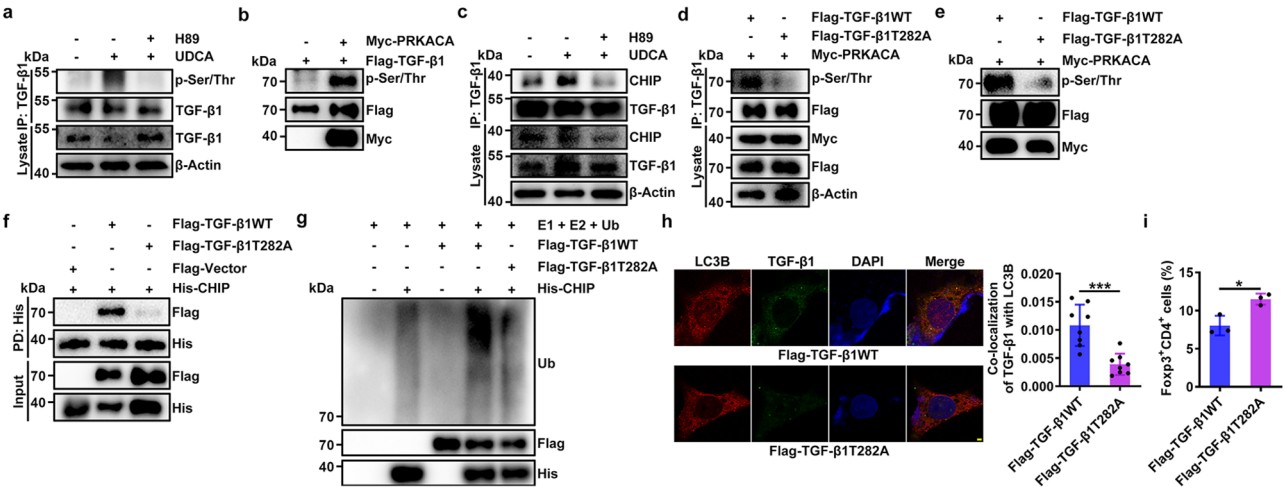

**Fig. 5 The binding of TGF-β1 phosphorylated at T282 by PKA to CHIP is increased. a** IB analysis of Ser/Thr-phosphorylated TGF-β1 in naïve CD4+ T cells stimulated with anti-CD3, anti-CD28 and 50 μM UDCA and treated with or without 2 μM H89 for 24 h. IB analysis was conducted after IP with anti-TGF-β1. **b** IB analysis of assay mixtures containing Flag-TGF-β1, Myc-PRKACA and ATP after reaction at 30 °C for 30 min in vitro. **c** IB analysis of CHIP and TGF-β1 in naïve CD4+ T cells stimulated with anti-CD3, anti-CD28 and 50 μM UDCA and treated with or without 2 μM H89 for 24 h. IB analysis was conducted after IP with anti-TGF-β1. **d** IB analysis of Ser/Thr-phosphorylated TGF-β1 in HEK293T cells transfected with vectors expressing Flag-TGF-β1WT, Flag-TGF-β1T282A, and Myc-PRKACA for 24 h. **e** IB analysis of assay mixtures containing Flag-TGF-β1 or Flag-TGF-β1T282A plus Myc-PRKACA and ATP after reaction at 30 °C for 30 min in vitro. **f** IB analysis of TGF-β1 in the mixture of His-CHIP and Flag-TGF-β1WT and the mixture of Flag-TGF-β1T282A after PD with anti-His magnetic beads. **g** IB analysis of TGF-β1 ubiquitination in the presence of E1, E2, Ub, purified CHIP and TGF-β1WT or TGF-β1T282A. **h, i** Immunofluorescence analysis of LC3B and TGF-β1 in HEK293T cells transfected with Flag-TGF-β1WT or Flag-TGF-β1T282A for 24 h. Scale bar, 2 μm. (**h**) FC analysis of Foxp3+CD4+ cells induced by culture with supernatants from these cells for 3 days (**i**). Representative results from two (**h**) or three (**a–g, i**) independent experiments are shown (n = 8 in **h**; n = 3 in **i**). *P < 0.05; **P < 0.01 (unpaired two-tailed Student's t test; mean and s.d.). See Source Data file for the exact P-values.

Then, we further verified the function of CHIP in tumor progression by establishing MC38 cells with CHIP knockout (MC38-*Chip*−/− cells) (Supplementary Fig. 8d). The levels of all TGF-β isoforms in the supernatant of MC38-*Chip*−/− cells were higher than those in the supernatant of MC38 cells (Fig. 6k), and there was no difference in the proliferation of MC38-*Chip*−/− and MC38 cells (Supplementary Fig. 8e). When inoculated into WT mice, MC38-*Chip*−/− tumors had an obvious growth advantage compared to MC38 tumors, but this effect was not observed in *Tgfbr2*f/f*Er-cre* mice (Fig. 6l, m). Before apparent differences in the size of the tumors were visible, higher TGF-β1, TGF-β2, and TGF-β3 levels were detected in the serum and tumor tissues of MC38-*Chip*−/− tumor-bearing mice (Fig. 6n, o). In addition, an increase in Treg cells was detected among the TILs of MC38-*Chip*−/− tumor-bearing mice (Fig. 6p). However, CHIP deficiency did not inhibit tumor progression in nude mice (Supplementary Fig. 8f). These results indicate that CHIP in tumors promotes antitumor T cell responses by suppressing TGF-β-mediated Treg cell differentiation.

CHIP deficiency also promotes the release of extracellular vesicles (EVs)[30], and tumor-derived EVs with TGF-β1 are conducive to Treg cell induction[31]. To further demonstrate the function of CHIP in tumor progression, we tested the effect of CHIP on the production of tumor-derived EVs. We isolated EVs from MC38 cells and MC38-*Chip*−/− cells (Supplementary Fig. 8g, h) and found that MC38-*Chip*−/− cells had a slightly enhanced ability for EV secretion (Supplementary Fig. 8i). Notably, EVs from MC38-*Chip*−/− cells contained more TGF-β1 (Supplementary Fig. 8j). Thus, tumors secrete more EVs, which have an increased TGF-β1 content, by downregulating CHIP, a mechanism that may also contribute to tumor immunosuppression.

**UDCA enhances the antitumor effects of anti-PD-1 therapy.**
Anti-PD-1 and anti-PD-L1 have distinct therapeutic effects on

tumors[32–34]. Anti-PD-1 and anti-PD-L1 block inhibitory signaling in T cells that is initiated by PD-L1 from the TME[35,36]. However, anti-PD-1 and anti-PD-L1 therapy theoretically cannot alleviate Treg-mediated tumor immunosuppression. We hypothesized that combination therapy with UDCA and either anti-PD-1 or anti-PD-L1 could overcome tumor immunosuppression in two ways, thereby significantly enhancing antitumor immunity: 1) protecting effector T cells from suppression by inhibiting Treg cell differentiation and activation and 2) preventing PD-1 signaling in effector T cells to suppress their activation (Supplementary Fig. 9a). In the clinic, UDCA is orally administered to dissolve cholesterol gallstones and treat primary biliary cholangitis[14]. Oral administration of Ursofalk (the trade name of UDCA) also inhibited B16-F10, LLC, and MC38 tumor growth (Fig. 7a). As expected, Ursofalk notably enhanced the efficacy of anti-PD-1 therapy against LLC and MC38 tumors, resulting in complete tumor eradication (Fig. 7a). Moreover, Ursofalk greatly improved the efficacy of anti-PD-1 therapy in anti-PD-1-insensitive B16-F10 tumors, leading to complete tumor regression in 28.6% of cases (Fig. 7a). After treatment with hydroxychloroquine (HCQ), an autophagy inhibitor, Ursofalk no longer enhanced anti-PD-1-mediated LLC tumor inhibition (Supplementary Fig. 9b). Moreover, Ursofalk did not improve the efficacy of anti-PD-1 therapy in LLC tumor-bearing mice with TGF-βRII deficiency or with Treg cell depletion (Supplementary Fig. 9c, d). Thus, the synergistic effects of UDCA with anti-PD-1 therapy are autophagy, TGF-β and Treg cell dependent.

Then, we assessed the UDCA effect on human anti-PD-1. When MC38 cells overexpressing human PD-L1 (MC38-huPD-L1) were introduced into *Pdcd1*em1(hPDCD1)/Smoc mice in which the mouse *Pdcd1* gene was replaced with the human *Pdcd1* gene (Supplementary Fig. 9e), combination treatment with Ursofalk and an anti-human PD-1 (SHR-1210) completely eradicated the established tumors (Fig. 7b). Analysis of TILs before tumor eradication indicated that combination treatment with Ursofalk

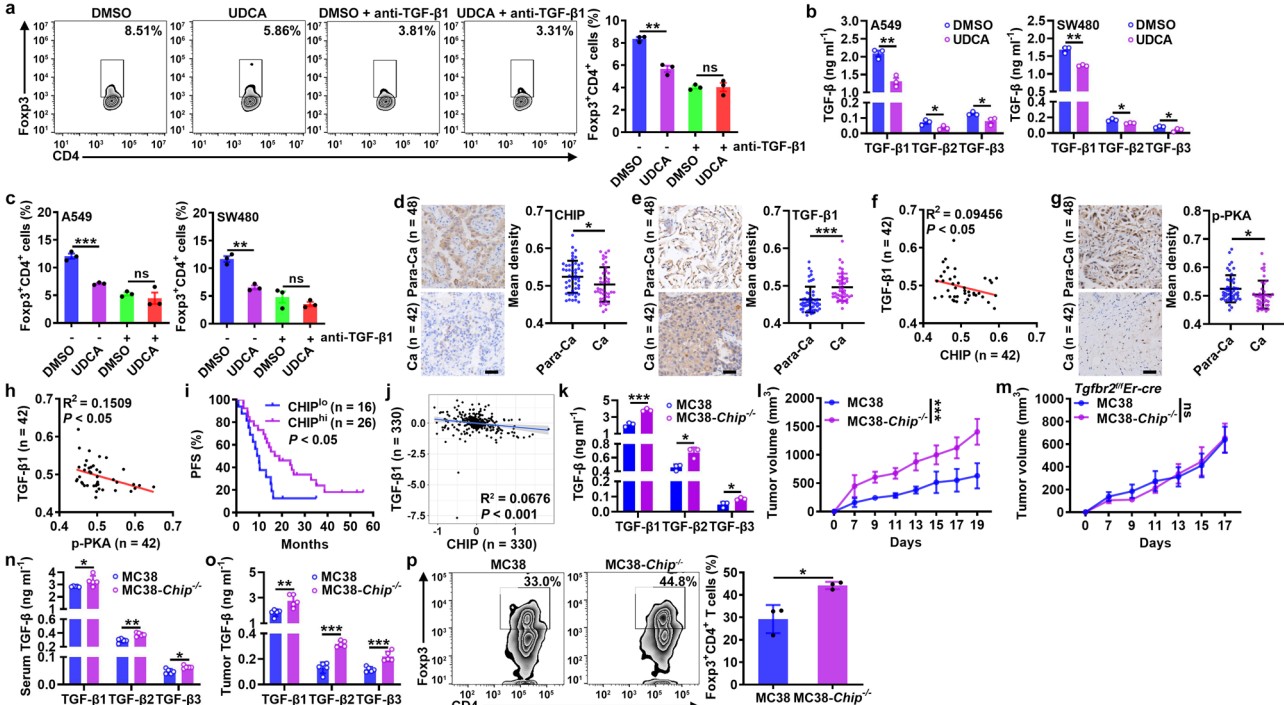

**Fig. 6 CHIP inhibits tumor progression by reducing the TGF-β level. a** FC analysis of Foxp3+CD4+ T cells among human naïve CD4+ T cells stimulated with anti-CD3, anti-CD28 and 50 μM UDCA with or without 2 μg ml⁻¹ anti-TGF-β1 for 4 days. **b** ELISA of TGF-β1, TGF-β2 and TGF-β3 proteins in supernatants of A549 and SW480 cells treated with 50 μM UDCA for 24 h. **c** FC analysis of Foxp3+CD4+ T cells among human naïve CD4+ T cells stimulated with anti-CD3 and anti-CD28 with or without 2 μg ml⁻¹ anti-TGF-β1 and cultured in the supernatants described in **b** for 4 days. **d–h** Immunohistochemical analysis of CHIP (**d**), TGF-β1 (**e**) and p-PKA (**g**) in Para-Ca tissues and Ca tissues (**d, e, g**) and correlations between CHIP and TGF-β1 (**f**), p-PKA and TGF-β1 (**h**) protein levels in Ca tissues (**f, h**) of NSCLC patients. **i** PFS curve for NSCLC patients with low versus high CHIP protein levels. **j** Correlation between CHIP and TGF-β1 protein levels in Ca tissues of HCC patients from the CPTAC database. **k** ELISA of TGF-β1, TGF-β2, and TGF-β3 proteins in the supernatants of MC38 and MC38-*Chip*⁻/⁻ cells. **l–p** Tumor sizes in MC38 and MC38-*Chip*⁻/⁻ tumor-bearing WT (**l**) and *Tgfbr2*ᶠ/ᶠ*Er-cre* (**m**) mice. ELISA of TGF-β1, TGF-β2 and TGF-β3 proteins in serum (**n**) and tumor tissues (**o**) and FC analysis of Foxp3+CD4+ T cells among TILs (**p**) from MC38 and MC38-*Chip*⁻/⁻ tumor-bearing mice on day 7 (**n, o**) and on day 19 (**p**). Representative results from three independent experiments are shown (*n* = 3 in (**a–c, k, p**); *n* = 5 in **l–o**). *$P < 0.05$; **$P < 0.01$; ***$P < 0.001$; ns, not significant (unpaired two-tailed Student's *t* test in (**a–e, g, k–p**); Spearman rank-order correlation test in **f, h, j**; log-rank test in **i**; mean and s.d.). See Source Data file for the exact *P*-values.

and SHR-1210 notably increased the IFN-γ⁺ CD8⁺ T cell component and decreased the Treg cells among TILs (Fig. 7c). Furthermore, combination treatment with Ursofalk and SHR-1210 significantly increased the levels of memory CD4⁺ and CD8⁺ T cells in peripheral blood, draining lymph nodes and the spleen (Fig. 7d, e and Supplementary Fig. 9f). Correspondingly, combination therapy protected mice against rechallenge with MC38-huPD-L1 but not B16-F10 tumors (Fig. 7f), implying the establishment of tumor-specific immune memory.

Finally, we retrospectively analyzed the survival of 211 patients with advanced NSCLC treated with single-agent anti-PD-1 or anti-PD-L1 therapy as second- or subsequent-line therapy. During the treatment period, 7 patients also received Ursofalk treatment (200 mg, bid) due to elevated bilirubin (Supplementary Table 2). Compared with the patients with anti-PD-1 or anti-PD-L1 treatment alone (median survival time, 3.28 months), these 7 patients had a favorable PFS (median survival time, 9.33 months) (Fig. 7g). Pancreatic cancer is insensitive to anti-PD-1 treatment[37,38]. There was 1 pancreatic cancer patient who received Ursofalk treatment during third-line anti-PD-1 therapy (Supplementary Table 2). In this patient, a notable decrease in tumor burden was observed (Supplementary Fig. 9g). These data collectively highlight the potential of UDCA in combination with anti-PD-1 or anti-PD-L1 therapy for stimulation of potent antitumor immunity.

## Discussion

TGF-β is essential for tumor immune evasion. However, the development of therapies based on inhibiting TGF-β signaling is slow[32]. The most extensively tested small molecule inhibitors target the ALK5 kinase. First-generation ALK5 inhibitors have failed due to considerable cardiac toxicity[39]. Although it does not exhibit discernable cardiac toxicity, the new ALK5 inhibitor galunisertib produced only modest therapeutic responses in pancreatic cancer and HCC in phase II trials[40–42]. There are also TGF-β blocking antibodies in phase I clinical trials[32]. However, as antibodies are macromolecules, assuring their tumor permeability is challenging. Here, we demonstrated that the clinical compound UDCA degraded both latent and activated TGF-β, as observed in various tumor cells and CD4⁺ T cells, indicating that UDCA is a pan-TGF-β inhibitor. Furthermore, UDCA exhibits excellent safety even for long-term application. Thus, UDCA is a ready-made TGF-β inhibitor with high efficacy and safety.

Similar to most cytokines, TGF-β is released in order to function. Canonically, secretory proteins are transported from the endoplasmic reticulum (ER) to the Golgi apparatus and subsequently to the plasma membrane via secretory vesicles[43]. Vesicular TGF-β cannot readily enter intracellular degradation pathways. Therefore, to date, little attention has been devoted to whether TGF-β undergoes degradation and how this degradation happens. For validation, we confirmed that TGF-β accumulated

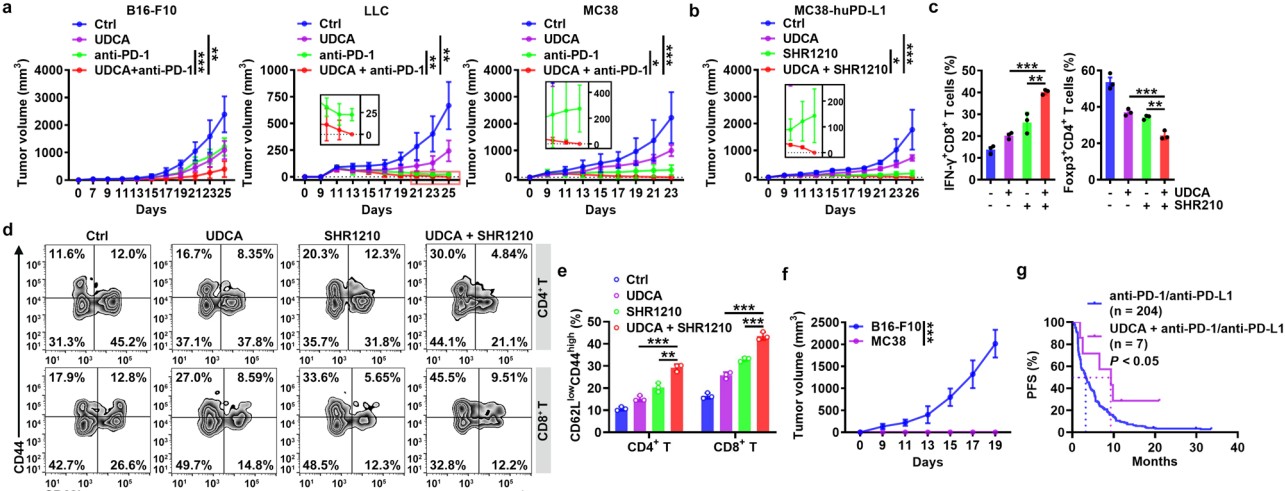

**Fig. 7 UDCA enhances the antitumor effects of anti-PD-1. a** Tumor sizes in B16-F10, LLC and MC38 tumor-bearing mice that received oral administration of 30 mg kg$^{-1}$ Ursofalk (UDCA) every other day along with i.p. injection of 50 μg anti-PD-1 every 4 days. **b–e** Tumor sizes (**b**) and FC analysis (on day 18) of IFN-γ+CD8+ T and Foxp3+CD4+ T cells among TILs (**c**); FC (**d**) and statistical (**e**) analyses (on day 33) of memory T cells in peripheral blood of MC38-huPD-L1 tumor-bearing mice that received oral administration of 30 mg kg$^{-1}$ UDCA every other day along with intravenous injection of 50 μg SHR1210 every 4 days. **f** Tumor sizes of tumor-free mice in (**b**) after rechallenge with B16-F10 or MC38-huPD-L1 cells. **g** PFS curve for NSCLC patients who received anti-PD-1 or anti-PD-L1 therapy with or without UDCA treatment. Representative results from three independent experiments are shown (mean and s.d.) ($n = 5$ in (**a, b, f**); $n = 3$ in (**c–e**). *$P < 0.05$; **$P < 0.01$; ***$P < 0.001$ (unpaired two-tailed Student's $t$ test except for log-rank test in (**g**); mean and s.d.). See Source Data file for the exact $P$-values.

in CD4+ T cells after treatment with autophagy inhibitors but not proteasome inhibitors, suggesting that its degradation is autophagy-dependent. Immunoelectron microscopy further confirmed the autophagosomal localization of TGF-β1. Then, we demonstrated that TGF-β was sorted into autophagosomes via p62. Furthermore, we verified that CHIP-mediated K63-linked ubiquitination of TGF-β is responsible for its autophagic sorting. However, we did not elucidate how TGF-β is sorted into autophagosomes in this study. Autophagy is initiated in the ER, and TGF-β may be sorted into autophagosomes after synthesis in the ER. In addition, free ribosomes involved in TGF-β synthesis might escape signal peptide-mediated ER targeting and directly release TGF-β into the cytoplasm for subsequent autophagic sorting, a possibility that requires further study.

Given the diverse functions of TGF-β, its degradation should be strictly regulated. We found that UDCA-mediated binding of TGF-β to TGR5 enhanced intracellular cAMP levels, resulting in the activation of PKA. Subsequently, PKA phosphorylated TGF-β at the T282 site, leading to the recruitment of CHIP. Then, CHIP ubiquitinated TGF-β, initiating its autophagic sorting and subsequent degradation (Supplementary Fig. 10). Therefore, an increase in TGF-β prevents PKA activation and reduces the CHIP protein level. Actually, both mechanisms were found to be utilized by tumors to prevent TGF-β degradation. In tumor tissues, the protein levels of phosphorylated PKA and CHIP were significantly decreased. Moreover, CHIP deficiency caused increases in the levels of TGF-β and Treg cells and accelerated tumor progression. Therefore, clarification of how PKA activation is inhibited and how CHIP is downregulated will provide strategies to decrease the TGF-β level in tumor patients, thus promoting the reestablishment of antitumor immunity.

UDCA has been reported to suppress acetyl-CoA carboxylase, which interferes with Treg cell induction[44,45]. However, acetyl-CoA carboxylase is unlikely to be involved in UDCA-mediated Treg cell regulation, because UDCA no longer induced Treg cells when TGF-β signaling was blocked. CHIP can decrease sensitivity to TGF-β signaling through ubiquitination and degradation of

Smad3[46]. Our results indicated that UDCA did not regulate the CHIP protein level, thereby altering the total protein level of Smad3. Therefore, UDCA is also unlikely to modulate TGF-β signaling through Smad3 degradation. However, we found that CHIP deficiency increased the secretion of tumor-derived EVs. Moreover, CHIP deficiency increased the TGF-β1 protein content in EVs, which probably stemmed from the increase in the total cellular TGF-β1 protein level. Given the immunosuppressive potential of TGF-β1-containing EVs[47], tumors may evade antitumor immunity by downregulating CHIP, leading to increased secretion of both TGF-β1 and TGF-β1-containing EVs.

Although anti-PD-1 and anti-PD-L1 therapies are effective in preventing the activation of T cell inhibitory signaling induced mainly by tumor PD-L1[48], theoretically, anti-PD-1 and anti-PD-L1 therapy cannot improve the immunosuppressive TME, in which TGF-β has a pivotal function. It was reported that treatment with a bifunctional fusion protein simultaneously targeting PD-L1 and TGF-β had stronger antitumor effects than treatment with either anti-PD-L1 antibodies or TGF-β traps alone[49]. Consistent with this study, our data demonstrated that combination treatment with UDCA and an anti-PD-1 notably increased antitumor CD8+ T cell responses, decreased the Treg cells among TILs, and enhanced tumor-specific immune memory, thus greatly improving the antitumor effects. In addition, the results of the retrospective analysis suggested the advantage of UDCA and anti-PD-1 combination therapy. Unlike bifunctional fusion proteins, UDCA is a safe clinical drug, which makes UDCA and anti-PD-1 combination therapy a more promising approach for tumor immunotherapy.

## Methods

**Human samples.** Human lung tissue samples from NSCLC patients and Para-Ca tissues and information on the 211 patients with advanced NSCLC treated with single-agent anti-PD-1 or anti-PD-L1 therapy were obtained from Zhejiang Cancer Hospital. The Zhejiang Cancer Hospital Ethics Committee approved the collection of human samples and patient information (IRB-2019-137). The Ethical Committee of Zhejiang University School of Medicine approved the collection of blood samples from healthy volunteers (2021-044). All the patients and healthy

volunteers were informed of the use of their samples, and signed consent forms were obtained.

**Mice and cell lines**. Female C57BL/6J mice and male BALB/c nude mice (6–8 weeks old) were purchased from Joint Ventures Sipper BK Experimental Animal Co. (Shanghai, China). NSG mice on a BALB/c background were purchased from Biocytogen (Beijing, China). $Tgfbr2^{f/f}$ mice on a C57BL/6J background were kindly provided by Prof. Bin Zhou (Center for Excellence in Molecular Cell Science, Shanghai, China). $Smad3^{-/-}$, OT-II, $Er$-$cre$, and $CD4$-$cre$ mice on a C57BL/6J background were purchased from the Jackson Laboratory (Farmington, CT, USA). $Foxp3^{GFP}$ knock-in mice on a C57BL/6J background were generously provided by Prof. Zhexiong Lian (South China University of Technology, Guangzhou, Guangdong, China). $Atg5^{f/f}$ mice on a C57BL/6J background were purchased from RIKEN BioResource Research Center (Tsukuba-shi, Japan). $Atg3^{f/f}$ mice on a C57BL/6J background were kindly provided by Dr. Wei Chen (Zhejiang University, Hangzhou, Zhejiang, China). $Atg3^{f/f}CD4$-$cre$ and $Atg5^{f/f}CD4$-$cre$ mice were obtained by crossing $Atg3^{f/f}$ or $Atg5^{f/f}$ mice with $CD4$-$cre$ mice. $Pdcd1^{em1(hPDCD1)/Smoc}$ mice on a C57BL/6J background were purchased from Nanjing Biomedical Research Institute of Nanjing University (Nanjing, Jiangsu, China). Mice were housed in a specific pathogen-free facility, and experimental protocols were approved by the Animal Care and Use Committee of Zhejiang University School of Medicine (ZJU20210045).

B16-F10, MC38, LLC, HEK293T, and NIH-3T3 cells were purchased from the American Type Culture Collection (ATCC; Manassas, VA, USA). A549 and SW480 cells were purchased from the Chinese Academy of Sciences (Shanghai, China). LLC-OVA cells were provided by Wei Yang (Southern Medical University, Guangzhou, Guangdong, China), and MC38-huPD-L1 cells were provided by Shanghai Hengrui Pharmaceutical Co., Ltd. (Shanghai, China). Murine MC38-$Chip^{-/-}$ cells were established in our laboratory. B16-F10, MC38, 293T, NIH-3T3, and SW480 cells were cultured in DMEM supplemented with 10% fetal bovine serum; LLC and A549 cells were maintained in RPMI 1640 medium supplemented with 10% fetal bovine serum. All cells were incubated at 37 °C in a humidified atmosphere of 5% $CO_2$-95% air.

**Antibodies**. Information on all the antibodies used in this study is provided in Supplementary Table 3.

**Tumor growth experiments**. Individual C57BL/6J mice were injected subcutaneously with $1 \times 10^6$ LLC cells, $2 \times 10^6$ MC38 cells, or $5 \times 10^5$ B16-F10 cells on day 0. Tumor-bearing mice received 30 mg kg$^{-1}$ UDCA (sc-222407, Santa Cruz Biotechnology, Santa Cruz, CA, USA) via i.p. injection every 2 days. Tumor size was monitored every other day by measurement with Vernier calipers. To silence TGF-βRII, $Tgfbr2^{f/f}Er$-$cre$ mice received tamoxifen (105-40-29-1, Sigma–Aldrich, St. Louis, MO, USA) in olive oil (8001-25-0, Sangon Biotech, Shanghai, China) (1 mg 100 μl$^{-1}$) via i.p. injection on days −10, −9, −8, 0, 7 and 14. In some experiments, nude mice, NSG mice, and $Tgfbr2^{f/f}Er$-$cre$ mice were subcutaneously injected with $1 \times 10^6$ LLC cells. In vivo depletion of Treg cells was achieved by injecting 100 μg of an anti-CD25 antibody (BE0012, Bio X Cell, West Lebanon, NH, USA) on days 7, 10, 13, and 17. To eliminate CD4$^+$ or CD8$^+$ T cells, individual mice received an anti-CD4 (BE0119, 120 μg) or anti-CD8 (BP0061, 40 μg) (Bio X Cell) antibody via i.p. injection on days 0, 3, 6, 9, 12 and 15. According to the criteria of the Animal Care and Use Committee of Zhejiang University School of Medicine, the maximal tumor size permitted was 8000 mm$^3$, tumor size in this study was not exceeded. TILs were prepared by enzymatic digestion with 1 mg ml$^{-1}$ collagenase IV (CLS-4, Worthington Biochemical Corp. Freehold, NJ, USA) and 0.5 mg ml$^{-1}$ DNase I (DN25, Sigma-Aldrich) at 37 °C for 90 min followed by Percoll (GE Healthcare, Uppsala, Sweden) gradient purification.

**In vitro culture of CD4$^+$ T cells**. CD4$^+$ CD62L$^+$ CD44$^-$CD25$^-$ naïve CD4$^+$ T cells were purified by FACS (BD FACSAria II, Becton, Dickinson and Company, Franklin Lake, NJ, USA) from splenocytes and peripheral lymph nodes and cultured at $0.4 \times 10^6$ cells per well in 96-well plates with plate-bound anti-CD3 (BE0001-1, 2 μg ml$^{-1}$) and anti-CD28 (BE0015-5, 2 μg ml$^{-1}$) (Bio X Cell) with or without latent TGF-β1 (299-LT) or TGF-β1 (240-B) (R&D, Minneapolis, MN, USA) stimulation at 37 °C. In some experiments, UDCA (U5127, 50 μM), TLCA (T7515, 50 μM), CDCA (C9377, 50 μM), LCA (L6250, 50 μM), DCA (D2510, 50 μM), GCA (G2878, 50 μM), CA (C1129, 50 μM) (all from Sigma-Aldrich), SBI-115 (S6806, 0.25 μM), H89 (S1582, 2 μM), IBMX (S5836, 100 μM), Baf-A1 (S1413, 1 nM or 10 nM), MG132 (S2619, 20 μM), INT747 (S7660, 100 μM), SB431542 (S1067, 0.5 μM), CHX (5 μM, S7418) (all from Selleck, Houston, TX, USA), MDL12330A (sc-201574, 5 μM, Santa Cruz Biotechnology), anti-TGF-β1 antibody (521707, 2 μg ml$^{-1}$, Biolegend, San Diego, CA, USA) and INT777 (HY-15677, 100 μM, MedChemExpress, Monmouth Junction, NJ, USA) were added at the initiation of T cell culture. In addition, mouse CD4$^+$ or CD8$^+$ T cells were isolated with a mouse CD4$^+$ T cell isolation kit (19852) or CD8$^+$ T cell isolation kit (19053) (StemCell, Vancouver, BC, Canada).

For culture of human in vitro CD4$^+$ T cells, naïve CD4$^+$ CD45RA$^+$ CD45RO$^-$ T cells were isolated from the population of peripheral blood mononuclear cells in healthy donors with a Human Naïve CD4$^+$ T Cell Isolation Kit II (17555, StemCell)

and stimulated with plate-bound anti-human CD3 (BE0001-2, 2 μg ml$^{-1}$) and anti-human CD28 (BE0248, 2 μg ml$^{-1}$) (Bio X Cell). In addition, human Treg cells or CD8$^+$ T cells were isolated with a Human Regulatory T Cell Enrichment Kit (19232) or Human CD8$^+$ T Cell Enrichment Kit (19053) (StemCell).

**Flow cytometry**. According to the manufacturer's instructions, intranuclear staining was carried out with fixation/permeabilization buffer solution (00-5123-43 and 00-5223-56, eBioscience, San Diego, CA, USA). For intracellular staining, cells were stimulated for 4 h at 37 °C in medium containing PMA (P1585, 50 ng ml$^{-1}$), ionomycin (I3909, 1 μg ml$^{-1}$) (both from Sigma–Aldrich), and brefeldin A solution (00-4506-51, eBioscience) and were then subjected to an intracellular staining protocol (00-8222-49, eBioscience). Stained cells were analyzed on a Beckman Coulter DxFLEX flow cytometer equipped with CytExpert experiment-based software (Beckman Coulter, Inc., 250 South Kraemer Boulevard Brea, CA, USA), and data were analyzed using FlowJo software (TreeStar, Ashland, OR, USA). The gating strategies are presented in Supplementary Fig. 11.

**Measurement of cytokine levels**. Tumor and cell culture supernatants and mouse and human serum were assayed by ELISA to measure the levels of TGF-β1 (88-8350-22, Invitrogen, Carlsbad, CA, USA), TGF-β2 (DY7346-05), and TGF-β3 (DY243) (R&D) according to the manufacturer's instructions.

**Treg suppression assay**. Suppression of CD4$^+$ or CD8$^+$ T cell proliferation by Treg cells was evaluated by CFSE labeling. Briefly, CFSE (C34570, Invitrogen)-labeled CD4$^+$ or CD8$^+$ T cells were seeded into a 96-well plate precoated with 2 μg ml$^{-1}$ anti-CD3 plus 2 μg ml$^{-1}$ anti-CD28 with Treg cells at a T cell:Treg cell ratio of 4:1. Five days later, cells were harvested, and the proliferation of CD4$^+$ or CD8$^+$ T cells was analyzed using a Beckman Coulter DxFLEX flow cytometer (Beckman Coulter, Inc.).

**Real-time PCR**. Total RNA was extracted using TRIzol reagent (9109, TaKaRa, Kusatsu, Shiga, Japan) and reverse transcribed into cDNA using a cDNA synthesis kit (RR047A, TaKaRa) according to the manufacturer's instructions. Real-time PCR was conducted using SYBR Green (RR420, TaKaRa). The following thermal cycling conditions were used for PCR: 1 cycle at 95 °C for 30 s followed by 40 cycles at 95 °C for 5 s and 60 °C for 34 s. Real-time PCR was performed with an Applied Biosystems 7500 Real-Time PCR System. The primer sequences are listed in Supplementary Table 4.

**Immunofluorescence and confocal microscopy**. CD4$^+$ T cells or NIH-3T3 cells subjected to different treatments were fixed with prewarmed 4% paraformaldehyde for 30 min and permeabilized with 0.1% Triton X-100 for 10 min. After blocking with 3% bovine serum albumin and 5% goat serum, the cells were incubated first with anti-TGF-β1 (3711, Cell Signaling, Danvers, MA, USA) and anti-LC3B (MA5-37852, Invitrogen) or anti-p62 (ab56416, Abcam, Cambridge, UK) antibodies at 4 °C overnight and then with DyLight 488- or DyLight 549-labeled secondary antibodies. Nuclei were stained with DAPI (D3571, Invitrogen). Stained tissue sections were viewed under an Olympus FluoView FV3000 confocal microscope and imaged using Olympus FluoView version 1.4a software (Olympus Corp, Tokyo, Japan). Images of cells and sections were acquired, and positively stained areas were analyzed.

**Immunogold electron microscopy**. TGF-β1-overexpressing HEK293T cells were stimulated with 10 nM Baf-A1 and 2.5 nM Torin1 for 2 h. Then, the cells were fixed with 4% paraformaldehyde and 0.1% glutaraldehyde in 0.1 M PBS (pH 7.4) overnight at 4 °C. The fixed samples were washed and incubated in 50 mM glycine for 5 min at room temperature (RT). The samples were permeabilized with 0.1% saponin for 40 min at RT. After being washed in PBS and blocked with 0.1% BSA for 30 min at RT, the samples were incubated first with anti-TGF-β1 antibodies overnight at 4 °C and then with mouse IgG conjugated to 1.4 nm gold particles overnight at 4 °C. The samples were then processed as follows: postfixed with 2.5% glutaraldehyde in 0.1 M PBS for 4 h at 4 °C; washed in PBS, water, and 0.02 M sodium citrate buffer (pH 7.0); treated with silver enhancement; and thoroughly washed in deionized water. The next day, the samples were washed three times in PBS for 10 min each and were then fixed with 1% osmium tetroxide for 60 min at RT. The samples were stained with 2% uranium acetate for 30 min after being washed three times in PBS. Then, the samples were dehydrated through a graded ethanol series (50%, 70%, 90% and 100%) and 100% acetone. Next, the samples were embedded in a 1:1 solution of Epon:acetone for 2 h at RT and a 3:1 solution of Epon:acetone overnight at RT. The next day, they were placed in fresh Epon for 8 h at 37 °C and were then embedded in Epon for 72 h at 65 °C. Thin sections were sliced, collected on grids and examined by EM (Tecnai G2 Spirit 120 kV, Thermo Fisher).

**Proximity ligation assay**. Purified naïve CD4$^+$ T cells were treated by NC siRNA or $p62$ siRNA for 24 h. Then, CD4$^+$ T cells were cultured in 48-well plates with plate-bound anti-CD3 and anti-CD28 for another 24 h. Cells were fixed with 4% paraformaldehyde for 15 min and permeabilized with 0.1% Triton X-100 for

10 min. After blocking with 3% bovine serum for 30 min, the cells were incubated with mouse anti-TGF-β1 and rabbit anti-LC3B antibodies overnight at 4 °C. Then, PLA was performed with Duolink In situ reagents (Sigma–Aldrich) according to the manufacturer's instructions. Then Samples were imaged using Olympus FluoView version 1.4a software (Olympus Corp). Images of cells and sections were acquired, and positively stained areas were analyzed.

**Plasmid and siRNA transfection**. HEK293T and NIH-3T3 cells were transfected with plasmids using JetPEI Transfection Reagent (Polyplus, New York, NY, USA) according to the manufacturer's protocol. CD4+ T and NIH-3T3 cells were transfected with scrambled NC or targeted siRNA using TransIT-TKO Transfection Reagent (Mirus Bio, Madison, WI, USA) according to the manufacturer's instructions. The PCMV6-TGF-β1-Myc-Flag (MR227339), PCMV6-CHIP-Myc-Flag (MR204258), PCMV6-PRKACA-Myc-Flag (MR205322) and PCMV6-p62-Myc-Flag (MR226105) plasmids were purchased from Origene Technologies (Rockwell, Maryland, USA). The PCMV6-TGF-β1-Flag, TGF-β1 mutant, PCMV6-CHIP-Flag, PCMV6-CHIP-His, PCMV6-PRKACA-Myc, pMX-PRKACA and PCMV6-p62_UBAΔ-Myc plasmids were constructed from the above plasmids. The siRNAs included sequences targeting murine *Tax1bp1*, *p62*, *Ndp52*, *Nbr1*, *Optineurin*, *Huwe1*, *Trim32*, *Hectd1*, *Ring1*, *Chip*, *Ranbp2*, and *Ubr5*, as well as NC siRNA (sc-154029, sc-29828, sc-141979, sc-149849, sc-39055, sc-61759, sc-61715, sc-145928, sc-38198, sc-44731, sc-36381, sc-143292 and sc-37007, respectively; Santa Cruz Biotechnology).

**Retroviral infection of CD4+ T cells**. Retroviruses were produced by transfecting Plat-E cells with 7.5 μg of pMX-*Ires-gfp* or pMX-*PRKACA-gfp*. The cell culture medium was replaced with fresh medium after 10 h, and the retrovirus-containing supernatant was collected after an additional 72 h. Naïve CD4+ T cells were first stimulated with anti-CD3 and anti-CD28 antibodies. At the 24- and 36-h time points, activated T cells were infected with 500 μl of the viral supernatant for 1 h by centrifugation at 1500 × g in the presence of 10 μg ml⁻¹ polybrene and incubated at 37 °C for an additional 1 h before removal from the viral supernatant and resuspension in the corresponding T cell medium for 4 days.

**IP and IB analyses**. Whole cells were washed twice with ice-cold PBS and lysed with cell lysis buffer (9803, Cell Signaling). Proteins in cell lysates (20–50 μg) were separated by sodium dodecyl sulfate (SDS)-polyacrylamide gel electrophoresis, transferred onto PVDF membranes (Millipore, Billerica, MA, USA), and probed with primary antibodies against the target proteins.

For complex co-IP, cell extracts were prepared by using lysis buffer [50 mM Tris (pH 7.4), 150 mM NaCl, 0.5% (vol/vol) Nonidet P-40, and 1 mM EDTA] supplemented with protease inhibitor cocktail. Lysates were incubated with antibody-coupled beads for 4–6 h at 4 °C. Immunoprecipitates were washed three times with the same buffer and subjected to IB analysis. Scanned images of uncropped blots scans are presented in the associated Source Data file.

**Mass spectrometry**. Each pulldown sample was run just in the separation gel and were cut into approximately 1-mm³ pieces, then subjected to in-gel trypsin digestion and dried. Samples were reconstituted in 5 μl of high-performance liquid chromatography solvent A (2.5% acetonitrile and 0.1% formic acid). A nanoscale reverse-phase high-performance liquid chromatography capillary column was created by packing 5-μm C18 spherical silica beads into a fused silica capillary (100-μm inner diameter × ~20-cm length) using a flame-drawn tip. After the column was equilibrated, each sample was loaded onto the column using an autosampler. A gradient was formed, and peptides were eluted with increasing concentrations of solvent B (97.5% acetonitrile and 0.1% formic acid). As the peptides eluted, they were subjected to MS as described above. MS analysis of the protein content and Ser/Thr phosphorylation of TGF-β1 was performed by using a Q Exactive system (Thermo Fisher). The results are presented in Supplementary Data 1.

**In vitro ubiquitination assay**. The in vitro ubiquitination assay was performed as described previously[50]. Briefly, Flag-TGF-β1WT, Flag-TGF-β1K315R, or Flag-TGF-β1T282A was transiently overexpressed with His-CHIP in HEK293T cells and purified using anti-Flag magnetic beads (M8823, Sigma–Aldrich) or anti-His magnetic beads (D291–11, Medical & Biological Laboratories, Tokyo, Japan) according to the manufacturer's protocol. Ubiquitination was analyzed with a ubiquitination kit (BML-UW9920, Enzo Life Sciences, Farmingdale, NY, USA) following protocols recommended by the manufacturer.

**In vitro kinase assay**. The in vitro kinase assay was performed as described previously[51]. Briefly, Flag-TGF-β1WT or Flag-TGF-β1T282A was transiently overexpressed with Myc-PRKACA in HEK293T cells and purified using anti-Flag magnetic beads or anti-Myc magnetic beads (88842, Thermo Fisher, Waltham, MA, USA) according to the manufacturer's protocol. The precipitated Flag-TGF-β1, Flag-TGF-β1T282A, and Myc-PRKACA proteins were resuspended in 40 μl of 1 × kinase buffer (9802) supplemented with 200 μM ATP (9804) (Cell Signaling). The reaction was carried out for 30 min at 30 °C and was terminated by the

addition of 20 μl 3 × SDS sample buffer. Each sample was then boiled for 10 min at 100 °C and subjected to SDS–PAGE and IB analysis using the indicated antibodies.

**Pulldown assay**. The pulldown assay was performed as described previously[52]. The Flag-TGF-β1WT or Flag-TGF-β1T282A fusion protein was expressed with His-CHIP in HEK293T cells and purified according to standard protocols. For the His pulldown assay, approximately 1 μg of His-CHIP bound to His beads was mixed with 1 μg of the Flag-TGF-β1WT or Flag-TGF-β1T282A fusion protein and incubated at 4 °C with gentle mixing. After overnight incubation, the beads were washed three times with cell lysis buffer, separated by SDS–PAGE with SDS sample buffer, and subjected to IB analysis.

**CRISPR-Cas9-mediated depletion of CHIP**. For depletion of CHIP in MC38 cells, the guide RNA plasmid (encoding the sgRNA) and Cas9-IRES-EGFP plasmid (encoding Cas9 and GFP) (Sigma–Aldrich) were cotransfected into MC38 cells. After cotransfection of the two plasmids into MC38 cells, GFP-positive cells were sorted using a Beckman Coulter DxFLEX flow cytometer (Beckman Coulter, Inc.). After sorting, single cells were cultured in 96-well plates. The CHIP knockout efficiency was confirmed by IB analysis. Selected MC38 cells with unchanged CHIP expression were used as control MC38 cells, while MC38 cells with CHIP deficiency were used as MC38-*Chip*⁻/⁻ cells. The sgRNA sequences are listed in Supplementary Table 4.

**EV isolation and characterization**. Fetal bovine serum was ultracentrifuged at 120,000 × g for 10 h to remove EVs and was then added to DMEM at a final concentration of 10% (v/v). MC38 and MC38-*Chip*⁻/⁻ cells were cultured in this medium to approximately 90% confluence in 10-cm cell culture dishes. Then, the medium was collected and centrifuged at 300 × g for 10 min at RT. All subsequent centrifugation steps were performed at 4 °C. Next, the supernatants were centrifuged at 2000 × g for 20 min and then at 10,000 × g for 30 min. Then, the supernatants were passed through 0.22 μm PVDF filters (Millipore) and subjected to ultracentrifugation at 120,000 × g for 70 min to precipitate EVs. The crude EV pellets were resuspended in a large volume of ice-cold PBS and were then subjected to ultracentrifugation at 120,000 × g for 70 min. The resulting pellets were resuspended in ice-cold PBS. The protein concentrations in EVs were measured using a BCA Protein Assay Kit (Thermo Fisher).

For electron microscopy detection, 200-mesh carbon grids were hydrophilized with a glow discharge instrument at 15 mA for 25 s. The EV solution was pipetted onto the 200-mesh carbon-coated copper grids and kept at RT for 1 min. After the excess suspension was removed with filter paper and the grid was washed twice with ddH₂O, the EVs were negatively stained with 2% uranyl acetate at RT for 1 min, and the excess suspension was removed and dried naturally. Images were acquired by electron microscopy (Tecnai G2 Spirit 120 kV, Thermo Fisher).

**The combination of in vivo tumor growth**. C57BL/6J mice and *Tgfbr2*ᶠ/ᶠ*Er-cre* mice were subcutaneously injected with 1 × 10⁶ LLC cells, 2 × 10⁶ MC38 cells, or 5 × 10⁵ B16-F10 cells on day 0. In addition, *Pdcd1*ᵉᵐ¹⁽ʰᴾᴰᶜᴰ¹⁾/*Smoc* mice were subcutaneously injected with 2 × 10⁶ MC38-huPD-L1 cells. Then, the tumor-bearing mice received UDCA (30 mg kg⁻¹, Santa Cruz Biotechnology) via i.p. injection every 2 days along with anti-mouse PD-1 (BE0273, 50 μg, Bio-X-cell) or anti-human PD-1 (SHR-1210, 50 μg, Hengrui Pharmaceutical Co., Ltd) via i.p. injection on days 7, 11, 15 and 19. In some experiments, mice received HCQ (60 mg kg⁻¹, HY-W031727, MedChemExpress) via i.p. injection every other day following tumor implantation. To test whether the combination of UDCA and anti-PD-1 therapy results in increases in memory CD4+ and CD8+ T cells, 2 × 10⁶ MC38 cells and 5 × 10⁵ B16-F10 cells were subcutaneously injected into *Pdcd1*ᵉᵐ¹⁽ʰᴾᴰᶜᴰ¹⁾/*Smoc* mice treated with UDCA and SHR-1210, in which MC38 tumors disappeared.

**Statistical analysis**. All statistical analyses were performed using GraphPad Prism 8.0 software. All data are expressed as the mean ± s.d. values. Unpaired Student's *t* test was used to compare differences between the two groups. One-way ANOVA followed by the Newman–Keuls test was used to compare differences among multiple groups. The log-rank test was used for survival analysis, and the Spearman rank-order correlation test was used for Pearson correlation analysis. A difference was considered significant if the *P*-value was < 0.05.

**Reporting summary**. Further information on research design is available in the Nature Research Reporting Summary linked to this article.

## Data availability

The MS data generated in this study have been deposited in the ProteomeXchange database under accession code PXD031801 (http://www.ebi.ac.uk/pride). All other data are available in the article and its Supplementary files or from the corresponding author upon reasonable request. Source data are provided with this paper.

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

## Acknowledgements

This work was supported by the National Natural Science Foundation of China (82130053, 31970845, 31770951, 31870876, and 82001662), Major Project of Hangzhou Health Science and Technology Plan (Z20200134) and Joint Preresearch Fund for Clinical Scientific Research of Hangzhou First People's Hospital Affiliated with Zhejiang University (YYJJ2019Z07). We thank Shuangshuang Liu in the Core Facilities, Zhejiang University School of Medicine, for technical support with the immunofluorescence analysis. We thank Chenyu Yang in the Center of Cryo-Electron Microscopy (CCEM), Zhejiang University, for her technical assistance with transmission electron microscopy. We thank the Key Laboratory of Immunity and Inflammatory Diseases of Zhejiang Province for the support.

## Author contributions

Y.Y.S., C.J.L., Z.B.S., C.X.Q., JO.L.W., J.B.C., C.Y.Z., X.C.Z., Z.Y.M., T.C., and J.F.G. performed various experiments; X.L. performed the MS analysis; A.F.L. discussed the manuscript. Z.J.C. and JN.L.W. designed the project and supervised the study; Z.J.C. and Y.Y.S. wrote the manuscript.

## Competing interests

The authors declare no competing interests.
