## [Peer Review File · Nature Communications]

Ursodeoxycholic acid reduces antitumor immunosuppression by inducing CHIP-mediated TGF- β degradationREVIEWER COMMENTS

Reviewer #1 (Remarks to the Author):

A potentially interesting study in that Ursodeoxycholic acid is being harnessed for cancer. However, for the most part the study appears to be a collection of somewhat disparate findings - some of which are extension from the lead authors prior publications e.g. on CHIP an E3 Ligase with a different mechanism of action. Abbreviations such as CHIP in the abstract do require to be corrected...

My comments are identical for editor and author:

In this study the authors examine the role of Ursodeoxycholic (UDCA) acid in disrupting TGF-beta driven tumorigenesis. The studies provide a complex array of mechanisms from UDCA-PKA driven phosphorylation of TGF-Beta, increasing binding and E3 ligase activity of CHIP (C-terminus of Hsc70-interacting protein) targeting TGF-beta, leading to its degradation and modulation of Treg activity that in turn modulates anti PD1 action. The study is interesting in that it addresses a pro-oncogenic role of TGF-beta and also harnesses UDCA, a well known therapeutic for cholestatic disorders such as Primary sclerosing cholangitis.

Major items that need to be addressed are as follows:

1. While these specific actions of CHIP and regulation of TGF-beta by UDCA have not been described before, multiple other mechanisms for UDCA actions are not addressed and would need to be (e.g. JBC 2014 Lee et al- DOI:<https://doi.org/10.1074/jbc.M113.491522>)
2. Likewise CHIP has been shown to regulate Smad3- which is downstream of TGF-Beta and this is not addressed, and would need to be (Xin et al J Biol Chem 2005 May 27;280(21):20842-50. doi: 10.1074/jbc.M412275200).
3. Figures are of uneven quality, small and difficult to follow, with statistical data not discernable.
4. The percentage of the HCCs (TCGA cohort of 372 cases) that have a high TGF-beta levels that correlate with CHIP expression are not provided and defining this new pathway of UDCA function as being relevant to human HCC would require interrogation all the multiple molecules being aligned as the study indicates using the TCGA database.
5. Clinical treatment strategies that could reflect these alterations in profiles are discussed and would need to be addressed more thoroughly in the context above. A model would be helpful.

Reviewer #2 (Remarks to the Author):

"Ursodeoxycholic acid breaks antitumor immunosuppression by inducing CHIP-mediated TGF- β Degradation" by Shen et al.

Major points:

- 1) Fig 1, The quality of the staining profile of Foxp3+ Tregs in the tumor tissues (Fig.1b, h) is too poor to make meaningful conclusions. Also it is strange that the frequency of Foxp3+ T regs in the tumor is so low (it is generally believed that the frequency of Tregs in tumor tissues is much higher than in the peripheral LNs and spleens 5-10%). Thus, the Foxp3 staining must be repeated and the quality be improved. Also, what is the level of classically studied activation markers of CD4/CD8 T cells (IFN γ , Perforin/Granzyme, TNF α , ki67, PD-1 etc)? Same question for Tregs (CD25, Ki67, St2, NLRP1).

- 2) Fig.1i and Suppl. Fig. 2f, if UDCA decreases TGF- β in Tregs, which might make sense to claim that UDCA downregulates Treg suppressive activity, as TGF- β is one of the possible factors to mediate Treg function to target responder T cells (although it is still not unanimous); however, how could the authors envision that the *Tgfr2* ko Tregs lose their suppression? Are the authors suggesting that Treg-derived TGF- β signals through their own receptors to confer Treg suppressive activity? If so, how could it happen? This type of experiments is based on the same number of Tregs added into the co-culture.
- 3) Fig. 2, again the quality of Foxp3 staining in this figure is also too poor to be convinced. The staining must be repeated to improve the quality. What type of the culture medium was used in this experiment? If the culture medium contains FBS, it should use serum-free medium (e.g. V-vivo 20) to exclude the UDCA effects on the endogenous TGF- β in the FBS and to further validate its function in CD4 T cell TGF- β .
- 4) Fig S3a, what is happening if you treat the cells with a low dose of TGF- β (like 0.2 ng/mL)? Same question in Foxp3-GFP T cells.
- 5) Fig S4f and h, the authors need to show the levels of Atg3 and Atg5 in the respective KO cells.
- 6) Fig S5A, what are the levels of P62?
- 7) Fig 3, the authors need to show that MG132 does not change the effect of UDCA on TGF- β levels.
- 8) Fig 3u, the authors need to confirm these data by doing co-IP experiments.
- 9) Fig4b-d, what if the effect of CHIP deletion on TGF- β localization of autophagic vesicles?
- 10) Fig4A, should not the levels of TGF- β be lower with UDCA treatment?
- 11) Fig 5, what is the effect of TGF- β mutant on the localization on autophagic vesicles, TGF levels and Treg generation?
- 12) The authors show that the levels of TGF- β 2 and 3 are also decreased by UDCA. Does a similar phenomenon happen with these 2 isoforms of TGF- β ?

Minor comments:

- "Introduction", "TGF- β is critical to induce the differentiation and proliferation of Treg" – this statement is incorrect, TGF- β can not induce the "proliferation" of Treg, the opposite is true.
- Fig S2a, is it the levels of LAP-TGF- β ? Or active TGF- β ?
- Fig S5i, we cannot see any dots on this figure.
- The discussion lacks a few references.
- The authors should provide the sequences of the plasmids

Reviewer #3 (Remarks to the Author):

This manuscript is a wonderful example of how exciting and relevant science can be. The authors have step by step discovered how an old drug can help to support modern treatment approaches - and span the gap between an original observation of UDCA to reduce tumor growth towards deciphering the underlying molecular pathway - actual showing that UDCA pushed endogenous TGF β 1 degradation - with all the consequences related to Treg formation. Fascinating how the authors discovered the autophagy pathway, the phosphorylation and ubiquitinylation sites of TGF β 1, and the underlying signaling axis. The preliminary clinical data are promising and in full support of all the mouse genetics and in vitro studies.

Maybe in line 4 above the Fig2 is "induced reduced" being a possible typo.
After Fig7a in the text: Moreover, ursofalk ... should be Ursolfalk

First of all, we would like to express our sincere gratitude to the reviewers for their constructive and positive comments. We have performed additional experiments and added new data in the revision. All the changes have been highlighted in yellow. Please see our point-by-point responses below.

To reviewer #1

A potentially interesting study in that Ursodeoxycholic acid is being harnessed for cancer. However, for the most part the study appears to be a collection of somewhat disparate findings - some of which are extension from the lead authors prior publications e.g. on CHIP an E3 Ligase with a different mechanism of action.

Abbreviations such as CHIP in the abstract do require to be corrected...

My comments are identical for editor and author:

In this study the authors examine the role of Ursodeoxycholic (UDCA) acid in disrupting TGF-beta driven tumorigenesis. The studies provide a complex array of mechanisms from UDCA-PKA driven phosphorylation of TGF-Beta, increasing binding and E3 ligase activity of CHIP (C-terminus of Hsc70-interacting protein) targeting TGF-beta, leading to its degradation and modulation of Treg activity that in turn modulates anti PD1 action. The study is interesting in that it addresses a pro-oncogenic role of TGF-beta and also harnesses UDCA, a well-known therapeutic for cholestatic disorders such as Primary sclerosing cholangitis.

Response: We appreciate very much for your positive comments on our study. According to your suggestions, we have added some new experiments and discussions, which will much improve the quality of our MS. In this study, we first found that UDCA showed antitumor effect by inducing Tregs, which is TGF- β dependent. Then, we found that UDCA could inhibit the TGF- β protein levels by autophagy-dependent degradation via p62-mediated autophagic sorting of TGF- β . The binding of p62 and substrate proteins requires the ubiquitination of these proteins. We did find the ubiquitination of TGF- β , so we further identified the E3 responsible for TGF- β

ubiquitination and found out CHIP. Given that PKA is also involved in UDCA-mediated TGF- β regulation, we further explored the role of PKA in this process and found that PKA-mediated phosphorylation is necessary for the binding of CHIP and TGF- β . Then, we analyzed the role of CHIP in tumor progress and its correlation with TGF- β . Finally, considering Tregs are critical for the formation of tumor immunosuppressive environment, we hypothesized that UDCA combined with anti-PD-1 could inhibit tumor both by attenuating tumor immunosuppressive environment and by preventing CD8 T cell from inactivation, and found that combined therapy greatly improved antitumor immunity. Therefore, our MS is not a collection of somewhat disparate findings, but logically organized. It should be pointed out that, the lead authors have no previous work related to CHIP and the uncovering of the novel function of CHIP is accidental. As you suggested, CHIP in the abstract and main text has been defined when it appeared for the first time. You can find our detailed point-to-point responses as follows:

Question 1: While these specific actions of CHIP and regulation of TGF-beta by UDCA have not been described before, multiple other mechanisms for UDCA actions are not addressed and would need to be (e.g. JBC 2014 Lee et al- DOI: <https://doi.org/10.1074/jbc.M113.491522>)

Response: UDCA is reported to suppress acetyl-CoA carboxylase which can restrain Treg development. However, acetyl-CoA carboxylase is unlikely to be involved in the Treg induction by UDCA. Because when TGF- β signaling was blocked, UDCA no longer induced Tregs in tumor mice. We have cited this literature and added the corresponding discussion in the revision highlighted in yellow (Page 20; Lines 397-399). In addition, CHIP abbreviation in the abstract and main text has been corrected.

Question 2: Likewise CHIP has been shown to regulate Smad3- which is downstream of TGF-Beta and this is not addressed, and would need to be (Xin et al J Biol Chem 2005 May 27;280(21):20842-50. doi: 10.1074/jbc.M412275200).

Response: CHIP decreases TGF- β signaling sensitivity through ubiquitination and degradation of Smad3. Our results showed that UDCA could not regulate CHIP protein levels which corresponds with the results that UDCA did not alter the total protein levels of Smad3. Thereby, UDCA is also unlikely to modulate TGF- β signaling through CHIP-mediated Smad3 degradation. We have cited this literature and added the corresponding discussion in the revision highlighted in yellow (Page 20; Lines 399-402). Moreover, CHIP is also reported to regulate the secretion of extracellular vesicles (EVs). Given the important role of EVs in tumor immunosuppression, we also detected the effect of CHIP on tumor EV secretion and found that CHIP deficiency increased not only the secretion of tumor EVs but TGF- β 1 proteins in EVs. This further supports the notion that CHIP is involved in tumor suppression. We have added these results in the revised Fig. S8g-j and the corresponding description in the revision highlighted in yellow (Page 16; Lines 309-315). We have also discussed about this highlighted in yellow (Page 20; Lines 402-406).

Question 3: Figures are of uneven quality, small and difficult to follow, with statistical data not discernable.

Response: We organized all the figures with high quality. When the figures were converted to PDF format, they were compressed which greatly impaired their quality. In the revision, all the figures are converted without compression by the submission system.

Question 4: The percentage of the HCCs (TCGA cohort of 372 cases) that have a high TGF-beta levels that correlate with CHIP expression are not provided and defining this new pathway of UDCA function as being relevant to human HCC would require interrogation all the multiple molecules being aligned as the study indicates using the TCGA database.

Response: It's a good suggestion. We also thought about it. However, TCGA database only provide the expression of gene levels including TGF- β and CHIP, so we could

not analyze the correlation between TGF- β and CHIP proteins based on the TCGA database. According to our results, CHIP regulates TGF- β at post-translational level. Therefore, we cannot provide the results as you suggested by TCGA database. However, we have analyzed the correlation between CHIP and TGF- β 1 protein levels in Ca tissues of HCC patients from CPTAC database and found that they are also negatively correlated which has been shown in revised Fig. 6j and the corresponding description has also been added in the revision highlighted in yellow (Page 15; Lines 293-295). However, we cannot provide the correlation between CHIP and p-PKA and the relationship of CHIP protein levels and patient survival time based on CPTAC database.

Question 5: Clinical treatment strategies that could reflect these alterations in profiles are discussed and would need to be addressed more thoroughly in the context above. A model would be helpful.

Response: We have provided a schematic to show how combinatorial therapy breaks tumor immunosuppression in revised Fig. S9a.

To reviewer #2

“Ursodeoxycholic acid breaks antitumor immunosuppression by inducing CHIP-mediated TGF- β Degradation” by Shen et al.

Response: We appreciate very much for your meaningful suggestions on our study, especially “addition of low concentration of TGF- β during Treg induction”. According to your suggestions, we have added some new experiments and discussions, which can further support our conclusions. You can find our detailed point-to-point responses as follows:

Question 1: Fig 1, The quality of the staining profile of Foxp3+ Tregs in the tumor tissues (Fig.1b, h) is too poor to make meaningful conclusions. Also it is strange that the frequency of Foxp3+ T regs in the tumor is so low (it is generally believed that the frequency of Tregs in tumor tissues is much higher than in the peripheral LNs and

spleens 5-10%). Thus, the Foxp3 staining must be repeated and the quality be improved. Also, what is the level of classically studied activation markers of CD4/CD8 T cells (IFN γ , Perforin/Granzyme, TNF α , Ki67, PD-1 etc)? Same question for Tregs (CD25, Ki67, St2, NLRP1).

Response: In previous revised Fig. 1b, h, we showed the frequency of Tregs in total CD45⁺ TILs, which is a little low. There will be some fluctuations in tumor size between different batches of tumor mice, which probably affects Treg frequency in TILs and caused the Tregs we detected were low. According to your comments, we have repeated Foxp3 staining and shown the frequency of CD4⁺CD25⁺Foxp3⁺ Tregs in CD4⁺ T cells of TILs in revised Fig. 1b, h and Fig. S1g. The frequency of Tregs and staining quality are both improved. We found UDCA notably reduced CD25⁺Foxp3⁺ Tregs of WT tumor mice but not *Tgfb2^{fl/fl}Er-cre* and *Smad3^{-/-}* tumor mice with TGF- β -signaling deficiency. We have also detected IFN- γ , Perforin, Granzyme, Ki-67 and PD-1 of CD4⁺ and CD8⁺ T cells and found that UDCA increased IFN- γ , Perforin, Granzyme and Ki-67 expression of both CD4⁺ and CD8⁺ T cells. In addition, UDCA decreased PD-1 expression on CD8⁺ T but not CD4⁺ T cells. Moreover, we found that UDCA hardly affected St2 on Tregs. These results are shown in revised Fig. 1b, S1b and the corresponding description has also been added and highlighted in yellow in the revision (Page 5-6; Lines 84-89). In previous Fig. S1e, we had detected Ki-67 in Tregs and found that UDCA did not affect it.

Question 2: Fig.1i and Suppl. Fig. 2f, if UDCA decreases TGF- β in Tregs, which might make sense to claim that UDCA downregulates Treg suppressive activity, as TGF- β is one of the possible factor to mediate Treg function to target responder T cells (although it is still not unanimous); however, how could the authors envision that the *Tgfb2* ko Tregs lose their suppression? Are the authors suggesting that Treg-derived TGF- β signals through their own receptors to confer Treg suppressive activity? If so, how could it happen? This type of experiments is based on the same number of Tregs added into the co-culture.

Response: In Fig. S2f, we did not claim that TGF- β RII-deficient Tregs lost their

suppression but found that UDCA did not affect the inhibitory capacity of Tregs with TGF- β RII deficiency. The previous publication has shown that Tregs with TGF- β R deficiency have defective immunosuppression (*Nat. Immunol.* 2008,9:632–640). So, in addition to as an effector molecule mediating suppressive function of Tregs, TGF- β probably affects Treg suppressive activity by acting on Tregs and inducing different intensity of TGF- β signaling via different concentration during their differentiation. Tregs from UDCA-treated mice with reduced TGF- β expressed lower levels of molecules related to their suppressive function, including TGF- β , CTLA4, ICOS and GITR. These results suggest that TGF- β is involved in the upregulation of TGF- β , CTLA4, ICOS and GITR, thus enhancing the suppressive activity of Tregs, which were inhibited in UDCA-treated tumor mice due to the reduction of TGF- β . Although UDCA still reduced TGF- β in *Tgfbr2^{f/f}Er-cre* tumor mice compared with that without UDCA treatment, TGF- β in *Tgfbr2^{f/f}Er-cre* tumor mice with or without UDCA treatment neither could act on T cells lacking TGF- β RII and affect Treg function. Thereby, there is no difference in the suppressive function of Tregs between these two groups of mice, as shown in Fig. S2f. Moreover, our results do not indicate that Treg-derived TGF- β signals through their own receptors to confer Treg suppressive activity. In tumor mice, the main source of TGF- β is from the tumor. Thus, it should be tumor TGF- β that binds to TGF- β R on T cells and affects Treg suppressive function. In Fig. S3o, p, we found that supernatant from UDCA-treated LLC-OVA with decreased TGF- β induced Tregs expressed reduced TGF- β , CTLA4, ICOS and GITR, and showed impaired function. These results further support that tumor TGF- β acts on T cells and affects suppressive activity of Tregs during their induction.

Question 3: Fig. 2, again the quality of Foxp3 staining in this figure is also too poor to be convinced. The staining must be repeated to improve the quality. What type of the culture medium was used in this experiment ? if the culture medium contains FBS, it should use serum-free medium (e.g. V-vivo 20) to exclude the UDCA effects on the endogenous TGF-b in the FBS and to further validate its function in CD4 T cell TGF-b.

Response: In previous Fig. 2 and Fig. S3a, we found that at high concentration of TGF- β cytokine, UDCA could not inhibit Treg differentiation. However, without TGF- β cytokine, UDCA significantly reduced the frequency of Foxp3⁺CD4⁺ T cells. These results suggest that TCR stimulation combined with endogenous TGF- β can drive Treg induction with low efficacy, which is inhibited by UDCA. Because UDCA can reduce endogenous TGF- β in CD4⁺ T cells. However, due to the substantial low production of endogenous TGF- β , Tregs induced by this way are very low. So, in the repeated experiments, we have also gotten the similar results. However, according to your suggestion in the next question, we have used different concentration of TGF- β cytokine to induce Tregs and detected the effect of UDCA on this induction. We found that if low dose of TGF- β cytokine was added, the induction efficacy of Tregs was improved and the inhibitory effect of UDCA on Treg differentiation could still observed. However, once high dose of TGF- β was added, this inhibitory effect of UDCA will be ignored and could not be observed. Therefore, it should be the TGF- β levels in the induction system but not the Foxp3 staining quality determined the low frequency of Foxp3⁺ Tregs. In the revision, we have replaced contour plot with dot plot in revised Fig. 2 and Fig. S3a to better demonstrate the inhibition of Treg induction by UDCA. In addition, each induction was followed by a Foxp3 qPCR result to further confirm the inhibition of Treg induction. We used V-vivo 20 medium to induced Tregs in previous Fig. S3m (revised Fig. S3n) to exclude the interference caused by FBS TGF- β and found UDCA still inhibited LLC-OVA-induced Tregs. According to your suggestion, we have also induced Tregs by CD4⁺ T cell endogenous TGF- β along with 0.2 ng/ml TGF- β 1 cytokine in V-vivo 20 medium and obtained the similar results. We have added these results in revised Fig. S3b and the corresponding description in the revision highlighted in yellow (Pages 7-8; Lines 130-132).

Question 4: Fig S3a, what is happening if you treat the cells with a low dose of TGF-b (like 0.2 ng/mL)? Same question in Foxp3-GFP T cells.

Response: It's a good suggestion. We have used different concentration of TGF- β 1

including 0.01, 0.2, 1 and 10 ng/ml to induce Tregs and detected the effect of UDCA on these induction. We found that exogenous TGF- β 1 dose-dependently promoted Treg induction. At low concentration of exogenous TGF- β 1 (no more than 0.2 ng/ml), endogenous TGF- β 1 synergistically induced Tregs with exogenous TGF- β 1, which was greatly inhibited by UDCA. When exogenous TGF- β 1 was sufficient (1 ng/ml or more) to induce Tregs, even if UDCA still degraded endogenous TGF- β 1, its inhibitory effect on Treg induction was no longer observed. We have substituted these results for previous Fig. S3a in the revision. We have also detected the effect of UDCA on Treg differentiation from naïve Foxp3-GFP CD4⁺ T cells in the presence of 0.2 ng/ml TGF- β 1 and found that higher Tregs could be detected, which could still be greatly inhibited by UDCA. These results have been shown in revised Fig. 2c. In addition, the corresponding description has also been added and highlighted in yellow in the revision (Page 7; Lines 123-127, 129-130).

Question 5: Fig S4f and h, the authors need to show the levels of Atg3 and Atg5 in the respective KO cells.

Response: The levels of Atg3 and Atg5 in the respective KO cells have been shown in revised Fig. S4f.

Question 6: Fig S5A, what are the levels of P62?

Response: We have added the p62 levels in revised Fig. S5a and found that p62 was accumulated overtime which was probably induced by the activation of TCR signaling. However, the p62 protein levels in CD4⁺ T cells with or without UDCA treatment showed no difference at the same time point. Therefore, UDCA did not affect the autophagic process.

Question 7: Fig 3, the authors need to show that MG132 does not change the effect of UDCA on TGF- β levels.

Response: We have shown this result in revised Fig. 3o and added the corresponding description in the revision highlighted in yellow (Page 10; Lines 185-186).

Question 8: Fig 3u, the authors need to confirm these data by doing co-IP experiments.

Response: We have confirmed that UDCA increased the co-localization of TGF- β 1 and p62 by co-IP experiments that were shown in revised Fig. 3w and the corresponding description has also been added in the revision highlighted in yellow (Page 11; Lines 211-212).

Question 9: Fig4b-d, what is the effect of CHIP deletion on TGF-b localization of autophagic vesicles?

Response: The autophagosome localization of TGF- β 1 was significantly reduced after CHIP silencing. We have added the result in revised Fig. 4e and the corresponding description in the revision highlighted in yellow (Page 12; Lines 235-236).

Question 10: Fig4A, should not the levels of TGF-b be lower with UDCA treatment?

Response: This experiment was performed in the presence of Baf-A1. We have added this information in the figure legend in the revision highlighted in yellow (Page 37; Line 824).

Question 11: Fig 5, what is the effect of TGF-b mutant on the localization on autophagic vesicles, TGF levels and Treg generation?

Response: TGF- β 1 mutant caused the increased TGF- β 1 in HEK293 cells, which had been shown Fig. S7i (Lane 1 vs. Lane 3). In the revision, we have also found that TGF- β 1 mutant caused the reduced autophagosome localization of TGF- β 1 and increased Treg generation. We have added these results in revised Fig. 5h, i and the corresponding description in the revision highlighted in yellow (Page 14; Lines 270-274).

Question 12: The authors show that the levels of TGF-b2 and 3 are also decreased by

UDCA. Does a similar phenomenon happens with these 2 isoforms of TGF-b?

Response: TGF- β 1 is the most abundant isoform, so we mainly explored the mechanisms responsible for UDCA-mediated TGF- β degradation by TGF- β 1. According to your comments, we have performed additional experiments about TGF- β 2 and TGF- β 3. We found that UDCA-mediated degradation of TGF- β 2 and TGF- β 3 was also Baf-A1 dependent. Moreover, we confirmed that UDCA-mediated degradation of TGF- β 2 and TGF- β 3 was CHIP dependent. These results suggest that similar phenomenon happens with TGF- β 2 and TGF- β 3. We have added the result in revised Fig. S4n, Fig. S6i and the corresponding description in the revision highlighted in yellow (Pages 10, 12; Lines 189-190, 238-239).

Question 13: “Introduction”, “ TGF-b is critical to induce the differentiation and proliferation of Treg” – this statement is incorrect, TGF-b can not induce the “proliferation” of Treg, the opposite is true.

Response: We apologize for the mistake and thank for your correction. In the revision we have modified this sentence (Page 4; Line 49).

Question 14: Fig S2a, is it the levels of LAP-TGF-b? Or active TGF-b?

Response: The clone number of the TGF- β antibodies we used is 1D11 that can detect TGF- β 1, 2, 3. The specification of 1D11 mentions “no cross-reactivity was observed with recombinant LAP in direct ELISAs”. Thereby, it can only detect the activated TGF- β and in Fig. S2a, it should be the levels of activated TGF- β . The antibody information was listed in Supplementary Table 4. Because it can detect all TGF- β isoforms, so we have changed TGF- β 1 to TGF- β in revised Fig. S2a, S3o and Fig. S8a.

Question 15: Fig S5i, we cannot see any dots on this figure.

Response: We organized all the figures with high quality. When the figures were converted to PDF format, they were compressed which greatly impaired their quality. In the revision, all the figures are converted without compression by the submission

system and you can see the dots in Fig. S5i.

Question 16: The discussion lacks a few references.

Response: We have cited some necessary references in the revision.

Question 17: The authors should provide the sequences of the plasmids.

Response: We have provided the sequences of the sgRNA and the information of plasmids used in this study was provide in the revision (Page 27; Lines 552-557).

To reviewer #3:

This manuscript is a wonderful example of how exciting and relevant science can be. The authors have step by step discovered how an old drug can help to support modern treatment approaches - and span the gap between an original observations of UDCA to reduce tumor growth towards deciphering the underlying molecular pathway - actual showing that UDCA pushed endogenous TGFbeta1 degradation - with all the consequences related to Treg formation. Fascinating how the authors discovered the autophagy pathway, the phosphorylation and ubiquitinylation sites of TGFbeta1, and the underlying signaling axis. The preliminary clinical data are promising and in full support of all the mouse genetics and in vitro studies. Maybe in line 4 above the Fig2 is "induced reduced" being a possible typo. After Fig7a in the text: Moreover, ursofalk ... should be Ursofalk

Response: First of all, we very appreciate for your positive comments on our work. In line 4 above the Fig2 "induced reduced" is not a typo. However, it might cause confused. So, we have changed the description to "supernatant from UDCA-treated LLC-OVA induced Tregs with reduced TGF- β 1, CTLA4, ICOS, and GITR" in the revision highlighted in yellow (Page 9; Lines 158-159). We have changed ursofalk to Ursofalk. Thanks again.

REVIEWERS' COMMENTS

Reviewer #1 (Remarks to the Author):

All concerns have been addressed.

Reviewer #2 (Remarks to the Author):

The authors have addressed my concerns and questions.

Reviewer #3 (Remarks to the Author):

Thank you; everything is clear now.

We would like to express our sincere gratitude to the reviewers for their constructive and positive comments again. Please see our point-by-point responses below.

To reviewer #1

Reviewer #1 (Remarks to the Author):

All concerns have been addressed.

Response: We greatly appreciate your work in the improvement of our MS. Thanks again!

Reviewer #2 (Remarks to the Author):

The authors have addressed my concerns and questions.

Response: We greatly appreciate your work in the improvement of our MS. Thanks again!

Reviewer #3 (Remarks to the Author):

Thank you; everything is clear now.

Response: We greatly appreciate your work in the improvement of our MS. Thanks again!